# Toward the Evaluation of Large Language Models Considering Score Variance across Instruction Templates

**Yusuke Sakai   Adam Nohejl   Jiangnan Hang   Hidetaka Kamigaito   Taro Watanabe**
Nara Institute of Science and Technology
{sakai.yusuke.sr9, nohejl.adam.mt3, hang.jiangnan.he1,
kamigaito.h, taro}@is.naist.jp

## Abstract

The natural language understanding (NLU) performance of large language models (LLMs) has been evaluated across various tasks and datasets. The existing evaluation methods, however, do not take into account the variance in scores due to differences in prompts, which leads to unfair evaluation and comparison of NLU performance. Moreover, evaluation designed for specific prompts is inappropriate for instruction tuning, which aims to perform well with any prompt. It is therefore necessary to find a way to measure NLU performance in a fair manner, considering score variance between different instruction templates. In this study, we provide English and Japanese crosslingual datasets for evaluating the NLU performance of LLMs, which include multiple instruction templates for fair evaluation of each task, along with regular expressions to constrain the output format. Furthermore, we propose the Sharpe score as an evaluation metric that takes into account the variance in scores between templates. Comprehensive analysis of English and Japanese LLMs reveals that the high variance among templates has a significant impact on the fair evaluation of LLMs.

## 1 Introduction

Decoder-based large language models (LLMs) have become foundational resources in the field of natural language processing, demonstrating superior natural language understanding (NLU) abilities and high pre-trained knowledge capacity in a wide variety of downstream tasks. Recently, LLMs can produce more human-like responses through instruction tuning (Wei et al., 2022a), which involves training the LLMs to respond appropriately to user instructions for various tasks.

Although LLM performance has been evaluated across various NLU tasks, the evaluation processes lack standardization in terms of prompts and output formats. This lack of standardization leads to differences in evaluation outcomes that cannot be attributed solely to the differences among LLMs. Moreover, the differences in prompts used for evaluation affect the evaluation results in NLU tasks (Zheng et al., 2023; Lu et al., 2022; Pezeshkpour and Hruschka, 2024; Zhao et al., 2021; Hou et al., 2024; Li et al., 2024; Sclar et al., 2024; Elazar et al., 2021; Madaan et al., 2024). In the specific case of instruction tuning, the goal is a prompt-independent generalization, though it is questionable to measure such generalization performance using prompts designed for specific targets.

For fair evaluation and comparison of the NLU performance of LLMs, we created benchmark datasets comprising multiple evaluation instruction templates for each NLU task based on the FLAN templates (Wei et al., 2022a), using five English NLU tasks and their corresponding Japanese tasks based on JGLUE (Kurihara et al., 2022). Additionally, we proposed a new evaluation metric, the Sharpe score, which accounts for the variance in LLM outputs due to template differences, inspired by the Sharpe ratio (Sharpe, 1966) used in finance to assess investment efficiency.

We demonstrated its effectiveness for the evaluation of template-based NLU capability, as well as for analysis of the NLU performance of multiple LLMs in various experimental scenarios, such as zero-shot versus fine-tuning settings and English versus Japanese settings. We examined how factors such as continuous training, instruction tuning, and language-specific knowledge affect knowledge-transfer capability. In order to enforce output generation in line with the expected response format, we accompanied each instruction template with a regular expression of the expected output for each task. The regular expressions are employed in constrained decoding methods as implemented in Outlines (Willard and Louf, 2023). We experimented with both constrained decoding and greedy decoding, demonstrating that constrained decoding with

regular expressions is effective for zero-shot evaluation. Our datasets and evaluation scripts are available at `https://github.com/naist-nlp/vite`.

## 2 Background and Related Work

The evaluation of the NLU capability of LLMs has mostly been based on benchmark datasets that combine several NLU tasks, such as GLUE (Wang et al., 2018) and SuperGLUE (Wang et al., 2019). Furthermore, NLU datasets that include domain-specific knowledge such as medical, economic, and mathematical knowledge (Jin et al., 2019; Baker et al., 2015; Pal et al., 2022; Shah et al., 2022; Chen et al., 2022; Amini et al., 2019; Hendrycks et al., 2021; Lin et al., 2022; Zhong et al., 2023b; bench authors, 2023; Suzgun et al., 2023; Liang et al., 2023) have been proposed for testing domain specific knowledge in LLMs. These benchmark datasets generally use automatic evaluation metrics, such as accuracy and F1 score.

These datasets are typically constructed in a concise format relevant to the particular task, providing only minimal information, such as questions and their answers. Therefore, the standard practices in evaluating LLMs employ instruction templates to make the datasets easy for LLMs to understand the instructions. The data instances are instantiated with instruction templates to yield natural language sentences, from which LLMs infer answers in an autoregressive manner.

Benchmark datasets for several languages other than English are available as well. Datasets for Japanese, a language we focus on in this study, include llm-jp-eval[1], JP Language Model Evaluation Harness[2], and Nejumi[3], all of which employ Japanese NLU datasets centered around JGLUE (Kurihara et al., 2022). The JP Language Model Evaluation Harness uses LLMs as classifiers by combining each question with corresponding answer choices and selecting the one with the lowest perplexity when all choices are ranked. Llm-jp-eval and Nejumi perform automatic evaluation by post-processing the generated text. In the evaluation used by Nejumi, if an answer cannot be obtained from the generated text, it assigns an arbitrary label, whereas the llm-jp-eval treats it as incorrect[4].

However, benchmarks for evaluating LLMs report results using only specific prompts, completely ignoring the performance variance of LLMs caused by different prompts. To mitigate the performance variance of LLMs due to different prompts, some LLMs such as FLAN (Wei et al., 2022a; Chung et al., 2024; Longpre et al., 2023), WizardLM (Xu et al., 2024), OpenAssistant (Köpf et al., 2023), and T0 (Sanh et al., 2022) enhance their generalization capabilities by instruction tuning with diverse templates, enabling robust responses to diverse inputs.

Prompt engineering (Wei et al., 2022b; Kojima et al., 2022; Zhong et al., 2023a; Yang et al., 2024; Zhou et al., 2023; Chen et al., 2024; Yao et al., 2023; Chen et al., 2023) has improved downstream task performance by converting input sentences into optimal prompts for LLMs. It focuses, however, on finding the best prompts for particular LLMs, making the engineered prompts unsuitable for evaluating the LLMs' NLU performance considering generalization capability.

While these approaches ensure the robustness of inputs, existing evaluation frameworks typically examine only a single template and ignore performance variance across multiple instruction templates. Consequently, to evaluate models' performance while taking into account their generalization ability, we need to find an evaluation method that incorporates variance across multiple instruction templates.

## 3 Evaluation Method

In our evaluation, we focus on the variance in results caused by differences in templates. To this end, we propose datasets and methods for evaluating the NLU performance of LLMs using multiple instruction templates. We evaluate performance in zero-shot and fine-tuning settings, but omit in-context learning, i.e., few-shot learning, settings. The prior studies (Mosbach et al., 2023; Zhang et al., 2024) have shown that the few-shot setting merely represents the exploration for optimal input prompts, capped by the performance of fine-tuning under the same number of examples.

### 3.1 Creation of Benchmark Datasets

As shown in Table 1, we employ five English NLU tasks and their corresponding Japanese tasks[5] to

---

[1] `https://github.com/llm-jp/llm-jp-eval`
[2] `https://github.com/Stability-AI/lm-evaluation-harness/tree/jp-stable`
[3] `https://wandb.me/nejumi`
[4] We confirmed the behavior in the source code.

[5] We selected tasks based on the JGLUE (Kurihara et al., 2022) datasets, excluding MARC (Keung et al., 2020) as it is currently unavailable. The JGLUE datasets were created from scratch based on the methodology used for the corresponding

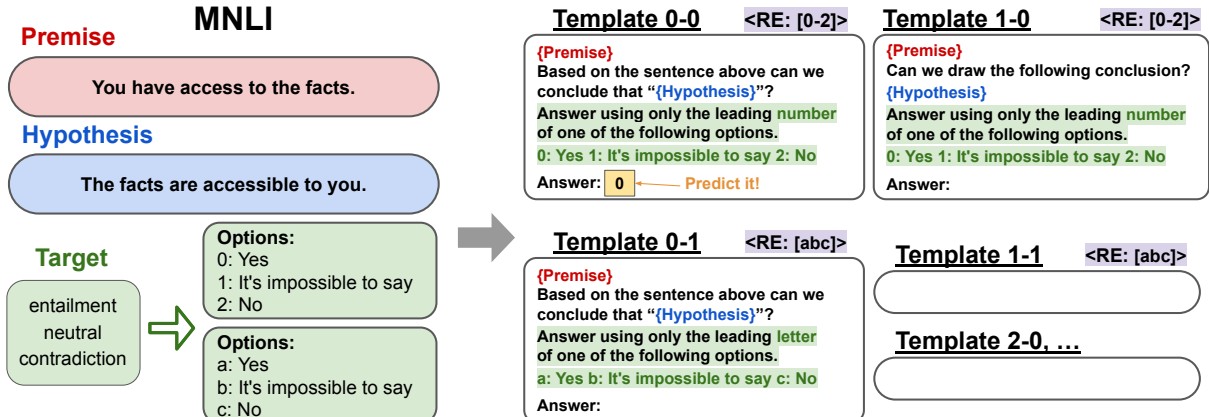

Figure 1: Examples of the dataset creation process for the MNLI task. We manually modified the original FLAN templates for evaluation, as highlighted in green. A regular expression (RE) shown in the purple area is attached to the expected answer format. We translated this template to create the Japanese templates described in Appendix E.

| Task | Lang. | #Templates | #Train | #Test |
|---|---|---|---|---|
| JCoLA (Someya et al., 2024) | Ja | 14 | 6,919 | 865 |
| CoLA (Warstadt et al., 2019) | En | 14 | 8,551 | 1,043 |
| JSTS (Kurihara et al., 2022) | Ja | 8 | 12,463 | 1,457 |
| STS-B (Cer et al., 2017) | En | 8 | 5,749 | 1,500 |
| JNLI (Kurihara et al., 2022) | Ja | 18 | 20,073 | 2,434 |
| MNLI (Williams et al., 2018) | En | 18 | 392,702 | 9,815 |
| JSQuAD (Kurihara et al., 2022) | Ja | 8 | 63,870 | 4475 |
| SQuAD (Rajpurkar et al., 2016) | En | 8 | 87,599 | 10,570 |
| JCSQA (Kurihara et al., 2022) | Ja | 12 | 8,939 | 1,119 |
| CSQA (Talmor et al., 2019) | En | 12 | 9,741 | 1,221 |

Table 1: Statistics of our datasets. The training and test datasets were constructed as Cartesian products of the templates, and the training and test instances, respectively. JCSQA represents JCommonsenseQA, and CSQA represents CommonsenseQA.

evaluate cross-lingual transfer capability and performance of multilingual LLMs. Appendix A provides details of each task and dataset.

We created the instruction templates for evaluation based on the FLAN templates (Wei et al., 2022a) by modifying them for the English tasks and then manually translating them into Japanese for the Japanese tasks. These instruction templates consist of structured prompts designed to guide the LLMs in performing specific tasks. Figure 1 shows examples of the dataset creation process for MNLI tasks. For each data instance, MNLI provides pairs of sentences, a premise, and a hypothesis. We then apply each instruction template to these sentence pairs to create natural language sentences to be used as input sequences. The expected output

English datasets, ensuring dataset alignment. Therefore, we can capture cross-lingual transfer performance that includes language-specific knowledge as noted by Sakai et al. (2024).

format for answers follows FLAN. We convert the answer labels to conversational text and instruct the LLMs to generate only the corresponding number or letter. We apply this procedure to other tasks to construct the entire benchmark dataset. All instruction templates are shown in Appendix F. Table 1 shows the number of templates and instances in the dataset. Furthermore, regular expressions for the expected answer format accompany each template, e.g., [0-2] in template 0-0 in Figure 1. By using regular expression-based constrained decoding methods, such as Guidance[6] or Outlines[7] (Willard and Louf, 2023), it is possible to ensure generation in the expected format without any post-processing. This allows the outputs to be used directly for evaluation, making the evaluation and comparison between LLMs fairer and simpler.

### 3.2 Experimental Settings

Table 2 shows the LLMs evaluated in our experiments. We report the results for both zero-shot and fine-tuning settings. For the fine-tuning setting, we use QLoRA (Dettmers et al., 2023)[8] to train the LLMs on each dataset. The detailed experimental settings of the parameters are described in Appendix B. We conduct greedy decoding and constrained decoding using regular expressions with Outlines (Willard and Louf, 2023). In greedy decoding, since the generated text may not follow the expected answer format, we referred to the post-processing method used by Ne-

---

[6] https://github.com/guidance-ai/guidance
[7] https://github.com/outlines-dev/outlines
[8] The performance differences between QLoRA and full fine-tuning are minimal (Dettmers et al., 2023; Liu et al., 2024; Dettmers and Zettlemoyer, 2023).

| LLMs | HuggingFace model name |
|---|---|
| *Japanese LLMs* | |
| OpenCALM-7B | cyberagent/open-calm-7b |
| StableLM-ja-7B | stabilityai/japanese-stablelm-base-alpha-7b |
| StableLM-ja-7B-inst | stabilityai/japanese-stablelm-instruct-alpha-7b |
| *English & Japanese LLMs* | |
| PLaMO-13B | pfnet/plamo-13b |
| Weblab-10B | matsuo-lab/weblab-10b |
| Weblab-10B-inst | matsuo-lab/weblab-10b-instruction-sft |
| LLM-jp-13B | llm-jp/llm-jp-13b-v1.0 |
| LLM-jp-13B-inst | llm-jp/llm-jp-13b-instruct-full-jaster-v1.0 |
| *Continuous English & Japanese LLMs* | |
| MPT-ja-7B | lightblue/japanese-mpt-7b |
| ELYZA-Llama-2-7B | elyza/ELYZA-japanese-Llama-2-7b |
| ELYZA-Llama-2-7B-inst | elyza/ELYZA-japanese-Llama-2-7b-instruct |
| *English LLMs* | |
| Llama-2-7B | meta-llama/Llama-2-7b-hf |
| Llama-2-7B-inst | meta-llama/Llama-2-7b-chat-hf |
| Llama-2-13B | meta-llama/Llama-2-13b-hf |
| Llama-2-13B-inst | meta-llama/Llama-2-13b-chat-hf |

Table 2: The LLMs used in our experiments and their corresponding model names on Hugging Face. Models with "inst" at the end of their names indicate that instruction tuning has been applied to them. The parameter count is also included in the model names. The classification of each model follows the claims of their creators. Japanese LLMs are trained mainly on Japanese pre-training data, English LLMs are trained mainly on English pre-training data, English & Japanese LLMs are trained on both English and Japanese pre-training data, Continuous English & Japanese LLMs are English pre-trained LLMs that are continuously trained on Japanese.

jumi[9]. We evaluate JCoLA and CoLA using accuracy (Acc) and the Matthews correlation coefficient (MCC) (Matthews, 1975); JSTS and STS-B using the Pearson and Spearman correlation coefficients; JNLI and MNLI using accuracy; JSQuAD and SQuAD using the exact match (EM) rate and F1 score; and JCommonsenseQA and CommonsenseQA using accuracy. These are the standard evaluation methods for each task.

The detailed post-processing methods and evaluation methods are described in Appendix B.

## 4 Experimental Results and Discussions

Results on the Japanese benchmark dataset are shown in Tables 3 and 4 for the zero-shot and fine-tuning setting, respectively. Similarly, results on the English benchmark dataset are shown in Tables 5 and 6 for the zero-shot and fine-tuning setting, respectively. Note that the results for the English benchmark dataset exclude the Japanese LLMs listed in Table 2. We will focus on important aspects in the following sections and defer more

---

[9] https://github.com/wandb/llm-jp

---

discussions to Appendix D.

### 4.1 Zero-Shot Setting

**Linguistic acceptability** In the JCoLA task in Table 3, even the best-performing LLM has accuracy equal to the chance rate, and MCC score is close to zero, indicating that none of the LLMs can perform the task successfully in the zero-shot setting. Table 5 shows the same tendency in the CoLA task, suggesting that linguistic acceptability judgment is a challenging task in the zero-shot setting. The low performance could be explained by the fact that JCoLA and CoLA employ answer labels annotated by linguists, in which their judgement might differ from non-experts in terms of acceptability since linguists prioritize grammaticality (Hu et al., 2023). Since LLMs are usually trained on general-domain corpora collected from the web, this difference may have an impact.

**Semantic textual similarity** In terms of zero-shot performance, shown in Table 5, Llama-2-13B-inst achieves high performance on the STS-B task in the English dataset. Furthermore, Table 3 shows that it also achieves high performance on the JSTS task in the Japanese dataset. This suggests that the LLM has a sufficient cross-lingual transfer capability for semantic textual similarity.

**Reading comprehension** From the JSQuAD task results shown in Table 3, the exact match rate improves after instruction tuning for Weblab-10B, LLM-jp-13B, ELYZA-Llama-2-7B, Llama-2-7B, and Llama-2-13B. However, no improvements are observed for StableLM-ja-7B after instruction tuning. This suggests that the quality of the instruction tuning data is important in the zero-shot setting.

**Commonsense reasoning** In CommonsenseQA and JCommonsenseQA results shown in Table 3 and Table 5, the Llama-2-7B-inst and Llama-2-13B-inst demonstrate a degree of language-transfer capability, although we would expect certain cultural differences embedded in commonsense knowledge of English and Japanese. However, if we focus at ELYZA-Llama-2-7B-inst[10], we observe a decrease in zero-shot performance compared to Llama-2-7B-inst. Nevertheless, in the results of the fine-tuning setting shown in Table 4, ELYZA-Llama-2-7B-inst scores improved compared to

---

[10] ELYZA-Llama-2-7B is continually trained from Llama-2-7B-inst, and ELYZA-Llama-2-7B-inst is instruction-tuned from ELYZA-Llama-2-7B.

| Model | JCoLA Acc/MCC | | JSTS Pearson/Spearman | | JNLI Acc | | JSQuAD EM/F1 | | JCommonsenseQA Acc | |
| | Greedy | Constrained | Greedy | Constrained | Greedy | Constrained | Greedy | Constrained | Greedy | Constrained |
|---|---|---|---|---|---|---|---|---|---|---|
| Chance Rate | **0.839**/0.000 | **0.839**/0.000 | 0.000/0.000 | 0.000/0.000 | 0.145 | 0.145 | 0.000/0.000 | 0.000/0.000 | 0.193 | 0.193 |
| OpenCALM-7B | **0.839**/0.000 | 0.838/0.009 | 0.052/0.051 | -0.010/-0.018 | 0.145 | 0.251 | 0.000/0.138 | 0.026/0.140 | 0.193 | 0.205 |
| StableLM-ja-7B | 0.594/-0.024 | 0.790/-0.006 | -0.026/-0.017 | -0.018/-0.019 | 0.261 | 0.209 | 0.227/0.401 | 0.165/0.333 | 0.205 | 0.204 |
| StableLM-ja-7B-inst | 0.541/-0.001 | 0.729/-0.014 | -0.026/-0.017 | -0.027/-0.028 | 0.281 | 0.259 | 0.214/0.398 | 0.175/0.354 | 0.205 | 0.205 |
| PLaMO-13B | 0.396/0.002 | 0.161/0.000 | -0.027/-0.025 | -0.032/-0.030 | **0.519** | 0.459 | 0.014/0.210 | 0.094/0.327 | 0.219 | 0.210 |
| Weblab-10B | **0.839**/0.000 | **0.839**/0.000 | 0.001/0.001 | -0.017/-0.014 | 0.145 | 0.218 | 0.001/0.267 | 0.099/0.262 | 0.193 | 0.215 |
| Weblab-10B-inst | **0.839**/0.000 | 0.600/-0.009 | 0.033/0.031 | 0.127/0.094 | 0.145 | **0.473** | 0.402/0.602 | 0.252/0.477 | 0.193 | 0.311 |
| LLM-jp-13B | 0.684/0.002 | **0.839**/0.000 | -0.052/-0.048 | 0.000/0.000 | 0.288 | 0.349 | 0.007/0.218 | 0.000/0.025 | 0.217 | 0.202 |
| LLM-jp-13B-inst | 0.500/-0.000 | **0.839**/0.000 | **0.585/0.572** | 0.000/0.000 | 0.445 | 0.225 | **0.857/0.923** | 0.000/0.022 | **0.783** | 0.202 |
| MPT-ja-7B | **0.839**/0.000 | 0.502/-0.001 | 0.023/0.016 | -0.016/-0.016 | 0.145 | 0.349 | 0.001/0.255 | 0.070/0.225 | 0.193 | 0.218 |
| ELYZA-Llama-2-7B | **0.839**/-0.004 | 0.827/**0.028** | 0.029/0.022 | 0.041/0.032 | 0.217 | 0.220 | 0.001/0.354 | 0.123/0.366 | 0.282 | 0.277 |
| ELYZA-Llama-2-7B-inst | 0.515/-0.001 | 0.500/-0.000 | 0.107/0.045 | 0.090/0.083 | 0.329 | 0.363 | 0.006/0.360 | **0.491/0.675** | 0.359 | 0.480 |
| Llama-2-7B | 0.589/0.004 | 0.426/-0.009 | 0.007/0.051 | 0.052/0.051 | 0.330 | 0.285 | 0.001/0.318 | 0.164/0.398 | 0.215 | 0.226 |
| Llama-2-7B-inst | 0.620/**0.006** | 0.187/0.020 | 0.007/-0.007 | 0.047/0.024 | 0.243 | 0.278 | 0.285/0.516 | 0.239/0.520 | 0.368 | 0.440 |
| Llama-2-13B | 0.675/0.005 | 0.549/0.002 | 0.089/0.088 | 0.013/0.011 | 0.214 | 0.200 | 0.001/0.312 | 0.151/0.368 | 0.250 | 0.237 |
| Llama-2-13B-inst | 0.679/0.000 | 0.473/0.004 | 0.217/0.236 | **0.312/0.286** | 0.181 | 0.174 | 0.310/0.528 | 0.176/0.540 | 0.385 | **0.540** |

Table 3: Results in the zero-shot setting on Japanese datasets. The bold font indicates the LLM with the highest evaluation performance for each task and decoding method, and the underline indicates the LLM with the second-highest evaluation performance. Chance Rate is the score when the LLM cannot infer anything and labels are assigned randomly. Note that LLM-jp-13B-inst includes some JGLUE tasks in its instruction-tuning data.

| Model | JCoLA Acc/MCC | | JSTS Pearson/Spearman | | JNLI Acc | | JSQuAD EM/F1 | | JCommonsenseQA MCC | |
| | Greedy | Constrained | Greedy | Constrained | Greedy | Constrained | Greedy | Constrained | Greedy | Constrained |
|---|---|---|---|---|---|---|---|---|---|---|
| OpenCALM-7B | 0.844/0.261 | 0.844/0.211 | 0.904/0.863 | 0.836/0.787 | 0.886 | 0.882 | 0.820/0.912 | 0.802/0.905 | 0.859 | 0.851 |
| StableLM-ja-7B | **0.859**/0.440 | 0.854/0.434 | 0.921/0.889 | **0.905/0.882** | 0.910 | 0.914 | 0.879/0.951 | 0.871/0.943 | 0.928 | 0.929 |
| StableLM-ja-7B-inst | 0.851/0.421 | 0.848/0.412 | 0.921/0.888 | 0.903/0.878 | 0.911 | 0.913 | 0.876/0.948 | 0.869/0.941 | **0.929** | **0.930** |
| PLaMO-13B | 0.838/0.376 | 0.837/0.371 | 0.919/0.884 | 0.897/0.869 | 0.912 | 0.912 | 0.882/0.949 | 0.852/0.938 | 0.917 | 0.916 |
| Weblab-10B | 0.856/**0.457** | 0.856/**0.456** | 0.910/0.871 | 0.897/0.857 | **0.919** | 0.919 | 0.888/0.954 | 0.884/0.948 | 0.894 | 0.895 |
| Weblab-10B-inst | 0.854/0.434 | 0.853/0.427 | 0.916/0.879 | 0.896/0.870 | 0.918 | 0.917 | 0.889/0.954 | 0.881/0.948 | 0.901 | 0.899 |
| LLM-jp-13B | 0.517/0.154 | 0.857/0.304 | **0.930/0.898** | 0.624/0.573 | 0.903 | 0.553 | **0.910/0.964** | 0.859/0.937 | 0.848 | 0.583 |
| LLM-jp-13B-inst | 0.519/0.146 | **0.861**/0.342 | **0.930/0.901** | -0.145/-0.070 | 0.882 | 0.533 | 0.906/0.963 | 0.870/0.941 | 0.885 | 0.846 |
| MPT-ja-7B | 0.852/0.401 | 0.854/0.392 | 0.919/0.885 | 0.902/0.876 | 0.914 | 0.913 | 0.002/0.468 | 0.885/0.951 | 0.891 | 0.891 |
| ELYZA-Llama-2-7B | 0.827/0.303 | 0.827/0.322 | 0.919/0.887 | 0.894/0.859 | 0.915 | 0.917 | 0.891/0.957 | 0.890/0.954 | 0.906 | 0.914 |
| ELYZA-Llama-2-7B-inst | 0.834/0.333 | 0.825/0.343 | 0.919/0.887 | 0.895/0.858 | 0.909 | 0.912 | 0.896/0.960 | 0.877/0.950 | 0.901 | 0.902 |
| Llama-2-7B | 0.812/0.302 | 0.817/0.324 | 0.913/0.879 | 0.893/0.869 | 0.910 | 0.912 | 0.891/0.957 | 0.879/0.949 | 0.857 | 0.859 |
| Llama-2-7B-inst | 0.810/0.245 | 0.793/0.244 | 0.910/0.876 | 0.889/0.865 | 0.900 | 0.905 | 0.892/0.959 | 0.883/0.950 | 0.836 | 0.844 |
| Llama-2-13B | 0.833/0.357 | 0.829/0.345 | 0.925/0.893 | 0.904/**0.882** | 0.917 | **0.921** | 0.901/0.962 | 0.889/0.954 | 0.893 | 0.894 |
| Llama-2-13B-inst | 0.818/0.301 | 0.823/0.329 | 0.914/0.877 | 0.894/0.868 | 0.893 | 0.915 | 0.898/0.962 | **0.891/0.956** | 0.878 | 0.886 |

Table 4: Results in fine-tuning setting on Japanese datasets.

Llama-2-7B-inst. This suggests that while the model has acquired knowledge through continuous training on Japanese data, it may have forgotten how to utilize it, leading to drop in accuracy in the zero-shot setting. At the same time, as shown in Table 5, ELYZA-Llama-2-7B and ELYZA-Llama-2-7B-inst achieve higher scores than Llama-2-7B in the zero-shot setting. This indicates that even with continuous training on Japanese data, the knowledge from the previous instruction tuning is preserved to some extent.

## 4.2 Fine-Tuning Settings

In the fine-tuning setting shown in Table 6, Llama2-13B is either the best or second-best model in most cases on the English dataset. Moreover, the pre-trained-only model achieves better results than its instruction-tuned version of Llama2-13B-inst. This demonstrates that instruction tuning does not guarantee better evaluation performance on the benchmark datasets, likely because instruction tuning aims to generalize the model for diverse queries.

As shown in Table 4, Llama2-13B achieves the highest or nearly the highest evaluation scores in JSTS, JNLI, and JSQuAD. In JCoLA, Weblab-10B achieves a particularly high score, and in JCommonsenseQA, StableLM-ja-7B-inst stands out with high scores. Comparison of these results with the results on English datasets suggests that LLMs can handle tasks such as JSTS, JNLI, and JSQuAD by leveraging their cross-lingual transfer capabilities. However, in the case of natural language inference (NLI, represented by MNLI and JNLI in our data), it has been pointed out that models might make

| Model | CoLA Acc/MCC | | STS-B Pearson/Spearman | | MNLI Acc | | SQuAD EM/F1 | | CommonsenseQA Acc | |
|---|---|---|---|---|---|---|---|---|---|---|
| | Greedy | Constrained | Greedy | Constrained | Greedy | Constrained | Greedy | Constrained | Greedy | Constrained |
| Chance Rate | **0.691**/0.000 | **0.691**/0.000 | 0.000/0.000 | 0.000/0.000 | 0.354 | 0.354 | 0.000/0.000 | 0.000/0.000 | 0.196 | 0.196 |
| PLaMO-13B | 0.556/-0.007 | 0.309/-0.001 | 0.021/0.025 | 0.005/0.007 | 0.335 | 0.339 | 0.002/0.242 | 0.019/0.333 | 0.194 | 0.202 |
| Weblab-10B | 0.676/0.009 | 0.629/0.003 | 0.065/0.066 | -0.025/-0.017 | 0.350 | 0.339 | 0.000/0.197 | 0.005/0.264 | 0.203 | 0.196 |
| Weblab-10B-inst | 0.653/-0.018 | 0.552/-0.015 | 0.000/0.000 | 0.430/0.440 | 0.350 | 0.338 | 0.000/0.001 | 0.000/0.001 | 0.202 | 0.211 |
| LLM-jp-13B | 0.682/0.012 | 0.500/0.000 | 0.000/0.026 | -0.001/-0.002 | 0.345 | 0.336 | 0.017/0.260 | 0.000/0.199 | 0.208 | 0.201 |
| LLM-jp-13B-inst | 0.500/-0.000 | 0.500/0.000 | **0.493**/**0.475** | 0.000/0.000 | 0.346 | 0.336 | **0.272**/0.696 | 0.000/0.214 | **0.435** | 0.201 |
| MPT-ja-7B | **0.691**/0.000 | 0.538/0.000 | 0.001/0.019 | 0.166/0.133 | 0.354 | 0.339 | 0.000/0.188 | 0.000/0.199 | 0.196 | 0.209 |
| ELYZA-Llama-2-7B | **0.691**/-0.018 | 0.500/0.001 | 0.182/0.179 | 0.267/0.245 | 0.353 | 0.344 | 0.000/0.228 | 0.014/0.258 | 0.266 | 0.237 |
| ELYZA-Llama-2-7B-inst | 0.523/0.005 | 0.517/0.009 | 0.173/0.158 | 0.086/0.066 | 0.341 | 0.353 | 0.001/0.231 | **0.199**/0.607 | 0.309 | 0.284 |
| Llama-2-7B | 0.681/-0.010 | 0.460/**0.042** | 0.181/0.181 | 0.132/0.131 | 0.353 | 0.341 | 0.000/0.229 | 0.023/0.301 | 0.216 | 0.207 |
| Llama-2-7B-inst | 0.517/0.002 | 0.488/0.001 | 0.182/0.157 | 0.184/0.142 | 0.343 | 0.346 | 0.236/0.677 | 0.147/0.703 | 0.348 | 0.420 |
| Llama-2-13B | **0.691**/0.010 | 0.582/0.040 | 0.076/0.078 | 0.064/0.064 | 0.362 | 0.354 | 0.000/0.210 | 0.066/0.397 | 0.251 | 0.219 |
| Llama-2-13B-inst | 0.572/**0.060** | 0.518/0.029 | 0.397/0.401 | **0.483**/**0.460** | **0.375** | **0.463** | 0.142/**0.731** | 0.115/**0.747** | 0.387 | **0.500** |

Table 5: Results in the zero-shot setting on English datasets.

| Model | CoLA Acc/MCC | | STS-B Pearson/Spearman | | MNLI Acc | | SQuAD EM/F1 | | CommonsenseQA Acc | |
|---|---|---|---|---|---|---|---|---|---|---|
| | Greedy | Constrained | Greedy | Constrained | Greedy | Constrained | Greedy | Constrained | Greedy | Constrained |
| PLaMO-13B | 0.842/0.628 | 0.842/0.629 | 0.901/0.902 | 0.877/0.877 | 0.842 | 0.840 | 0.757/0.912 | 0.740/0.907 | 0.729 | 0.728 |
| Weblab-10B | 0.843/0.625 | 0.842/0.623 | 0.896/0.898 | 0.885/0.886 | 0.836 | 0.837 | 0.764/0.913 | 0.736/0.903 | 0.657 | 0.657 |
| Weblab-10B-inst | 0.835/0.605 | 0.833/0.600 | 0.908/0.906 | 0.908/0.908 | 0.843 | 0.844 | 0.763/0.915 | 0.736/0.905 | 0.665 | 0.664 |
| LLM-jp-13B | 0.576/0.268 | 0.766/0.403 | 0.897/0.899 | 0.627/0.688 | 0.705 | 0.603 | 0.443/0.793 | 0.760/0.915 | 0.658 | 0.456 |
| LLM-jp-13B-inst | 0.576/0.271 | 0.769/0.410 | 0.912/0.911 | 0.883/0.887 | 0.713 | 0.576 | 0.413/0.781 | 0.756/0.914 | 0.677 | 0.487 |
| MPT-ja-7B | 0.816/0.559 | 0.815/0.555 | 0.902/0.902 | 0.888/0.890 | 0.837 | 0.801 | 0.000/0.493 | 0.733/0.903 | 0.702 | 0.701 |
| ELYZA-Llama-2-7B | 0.853/0.654 | 0.853/0.654 | 0.910/0.911 | 0.916/**0.918** | 0.875 | 0.867 | 0.793/0.931 | 0.771/0.925 | 0.757 | 0.760 |
| ELYZA-Llama-2-7B-inst | 0.857/0.665 | 0.862/**0.679** | 0.911/0.911 | **0.918**/**0.918** | 0.875 | 0.876 | 0.791/0.930 | 0.768/0.923 | 0.751 | 0.754 |
| Llama-2-7B | 0.858/0.661 | 0.859/0.665 | 0.908/0.910 | 0.898/0.902 | 0.877 | 0.881 | 0.795/0.933 | 0.773/0.925 | 0.770 | 0.770 |
| Llama-2-7B-inst | 0.855/0.656 | 0.850/0.646 | **0.917**/**0.917** | 0.895/0.899 | 0.877 | 0.880 | 0.798/0.933 | 0.775/0.925 | 0.758 | 0.764 |
| Llama-2-13B | **0.871**/**0.693** | **0.863**/0.678 | 0.913/0.914 | 0.904/0.906 | 0.888 | **0.893** | **0.802**/**0.938** | **0.787**/**0.934** | **0.804** | **0.799** |
| Llama-2-13B-inst | 0.847/0.641 | 0.854/0.657 | 0.916/0.915 | 0.902/0.904 | **0.889** | 0.892 | 0.798/0.936 | **0.787**/0.933 | 0.786 | 0.796 |

Table 6: Results in the fine-tuning setting on English datasets.

predictions based solely on superficial features due to overfitting (Kavumba et al., 2022; McCoy et al., 2019; Wang et al., 2022; Tang et al., 2023; Du et al., 2023). Thus, further investigation is necessary to justify whether these results are truly due to cross-lingual transfer, or not.

In JCoLA and JCommonsenseQA, ELYZA-Llama-2-7B-inst, which is the continuously trained model from Llama-2-7B-inst, achieves higher scores compared to Llama-2-7B-inst in both accuracy and MCC in JCoLA, as well as improved scores in JCommonsenseQA. This suggests that continuous training with Japanese data contributes to improvement in language acceptability tasks and commonsense reasoning tasks, and cross-lingual transfer through continuous training is effective.

As we can see in Table 4 and Table 6, the scores in JCoLA and CoLA decrease after instruction tuning for some LLMs. One possible factor is that instruction tuning involves training to improve the models' ability to respond to diverse inputs, enabling them to accept even linguistically incorrect input sentences. As a result, the instruction-tuned LLMs may have become more lenient in their judg-ment of acceptability, leading to errors in this task.

### 4.3 Decoding Methods

In the zero-shot setting shown in Table 3 and Table 5, constrained decoding with regular expressions generally achieves higher performance than greedy decoding. However, in the fine-tuning setting shown in Table 4 and Table 6, greedy decoding generally achieves higher performance than constrained decoding. Therefore, especially when evaluating the zero-shot setting, it is reasonable to use constrained decoding to eliminate errors due to differences in output formats.

Additionally, in Table 3, we can see that LLM-jp-13B-inst shows a significant difference in scores between greedy and constrained decoding. One possible reason for this is the influence of the instruction data, specifically the Jaster[11] dataset created, which is based on the JGLUE datasets. We hypothesize that due to instruction tuning with Jaster, higher generation probabilities are assigned to certain words, which may have worked well with greedy decoding but not with constrained decoding

[11] https://github.com/llm-jp/llm-jp-eval

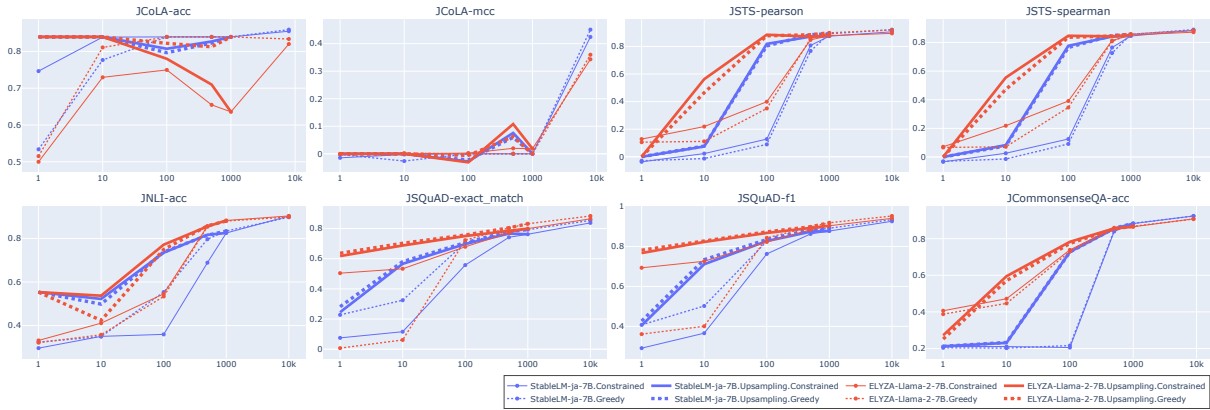

Figure 2: The number of sentences used in fine-tuning and the evaluation scores for each task. Thin lines represent training for one epoch, while thick lines represent training by upsampling to achieve a total of 1000 sentences. We use StableLM-ja-7B-inst and ELYZA-Llama-2-7B-inst.

([Jain et al., 2024](#) makes a similar observation about instruction tuning).

### 4.4 How Many Examples Are Required for Adequate Evaluation in the Fine-Tuning Setting?

We investigated the number of sentences required for the fine-tuning setting to evaluate the NLU performance of LLMs. Figure 2 shows the evaluation scores when fine-tuning StableLM-ja-7B-inst and ELYZA-Llama-2-7B-inst with 1, 10, 100, 500, 1000, and 10000 sentences[12]. Thin lines represent the results of fine-tuning with each number of sentences only once, while thick lines represent the results of repeated fine-tuning with the respective number of sentences to achieve a total of 1000 sentences, e.g., training with 10 sentences 100 times or with 500 sentences 2 times to achieve a total of 1000 sentences.

Figure 2 shows that for the four datasets other than JCoLA, the difference in evaluation scores between training with 1000 and 10000 sentences is only marginal. Furthermore, for JSTS, training with 100 sentences repeated 10 times achieves sufficient inference accuracy. For JSQuAD, repeated training with a small number of sentences, such as 1 or 10, improves evaluation scores.

The reason why JCoLA does not show the same tendency as the other datasets is unclear. It may be due to the difficulty of the task itself or due to the complexity of the dataset. In conclusion, to adapt the output, we only need to train with a small number of examples. Around 1000 sentences

are generally sufficient to fine-tune the model adequately for evaluation of its NLU capabilities.

## 5 Analysis Considering Variance Among Templates

### 5.1 Necessity of Evaluation Using Multiple Templates

Figure 3 shows the evaluation results in the fine-tuning setting with only a single template on the Japanese dataset. The accuracy of each template varies greatly for JNLI and JCommonsenseQA, depending on whether the template's answer format uses letters or numbers. Moreover, in JSTS and JCoLA, certain templates result in lower scores. On the other hand, when constrained decoding is applied, some models and tasks produce more stable outputs. This suggests that while the models can respond to the input sentences, they fail to faithfully follow the correct output format. In other words, although we can observe generalization to some extent when a model is fine-tuned with a single template, the performance often varies due to a mismatch between the trained template and the answer format expected at inference time. Evaluation using a single template should, therefore, be avoided. It is instead necessary to use multiple templates for evaluation and to assess the variance among them in order to measure the generalization performance properly. This finding also confirms the results of studies that employed multiple templates for training ([Wei et al., 2022a](#); [Xu et al., 2024](#); [Köpf et al., 2023](#); [Sanh et al., 2022](#)), suggesting that model generalization and its language transfer performance improve by exposing the model to diverse input formats through the use of multiple templates.

---

[12]For JCoLA and JCommonsenseQA, as the training data is less than 10000 sentences, we report the results using all available training data instead.

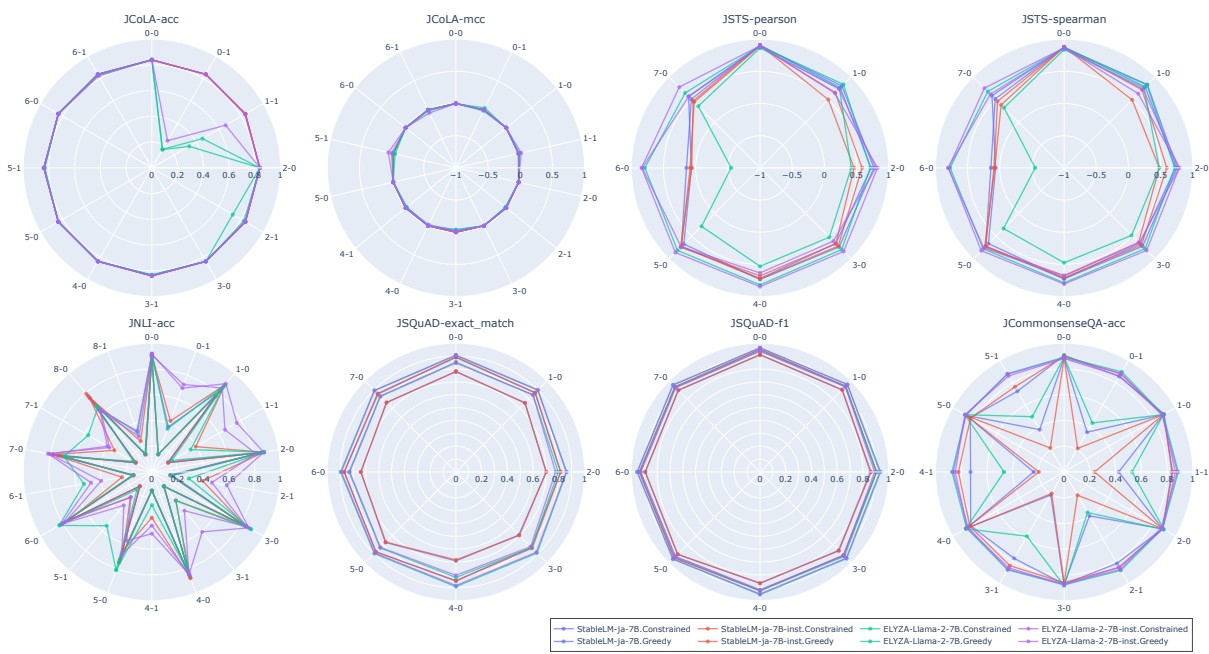

Figure 3: Evaluation results for each template when trained with only a single template. The results show the evaluation for each template after training only using the template with ID 0-0 (positioned at the top in the figure). The first part of the template number indicates the type of template, and the second part indicates the type of answer format. The types of answer formats are described in Figure 1. The LLMs used for evaluation are StableLM-ja-7B, StableLM-ja-7B-inst, ELYZA-Llama-2-7B, and ELYZA-Llama-2-7B-inst.

## 5.2 Evaluation Metrics Considering Performance Variance Among Templates

**Sharpe score** LLMs are expected to provide correct answers to diverse prompts, rather than only responding to specific prompts. Therefore, we propose the Sharpe score, an evaluation metric designed to evaluate both the robustness and accuracy of outputs by considering different instruction templates. The Sharpe score is based on the Sharpe ratio (Sharpe, 1966), which is used in finance to assess investment efficiency. The Sharpe ratio is used as a measure of the risk-adjusted return of an investment. The Sharpe ratio can be expressed as follows:

$$Sharpe\ ratio = \frac{R_p - R_f}{\sigma_p}, \qquad (1)$$

where $R_p$ is the return of the portfolio, $R_f$ is the risk-free rate, and $\sigma_p$ is the standard deviation of the portfolio return.

When applying this concept to our evaluation, the return of the portfolio $R_p$ corresponds to the average of the evaluation scores $\mu_{score}$, the risk-free rate $R_f$ corresponds to the chance rate, and the standard deviation of the portfolio return $\sigma_p$ corresponds to the standard deviation of the evaluation scores for each template $\sigma_{score}$. Since the chance rate is constant for each task, we can ignore it.

We define the Sharpe score as follows:

$$Sharpe\ score = \frac{\mu_{score}}{\alpha\sigma_{score} + 1}, \qquad (2)$$

where $\alpha$ is a parameter that controls the impact of variance in scores among templates. We add 1 to the denominator as a smoothing term to avoid the zero-division issue. When $\alpha$ is 0, the score is reduced to an average of performance evaluation metrics. When $\alpha$ is 1, the Sharpe score is computed analogously to the Sharpe ratio. For values greater than 1, the variance in results across templates leads to a proportionally larger penalty. The default parameter of $\alpha$ is set to 1.0. The Sharpe score can be applied to any evaluation metric as it adjusts based on the average result while considering variance. The more detail experimental results with the Sharpe score are discussed in Appendix C.

**Ranking** Figure 4 shows the changes in the rankings among the models, using the Sharpe score by incrementing the hyperparameter $\alpha$ from 0 to 2 by steps of 0.1 in the Japanese dataset. Appendix C shows the results for the English dataset sharing a similar tendency. While the mean and variance values are constant for each model, the change in the hyperparameter $\alpha$ reflects the degree of impact

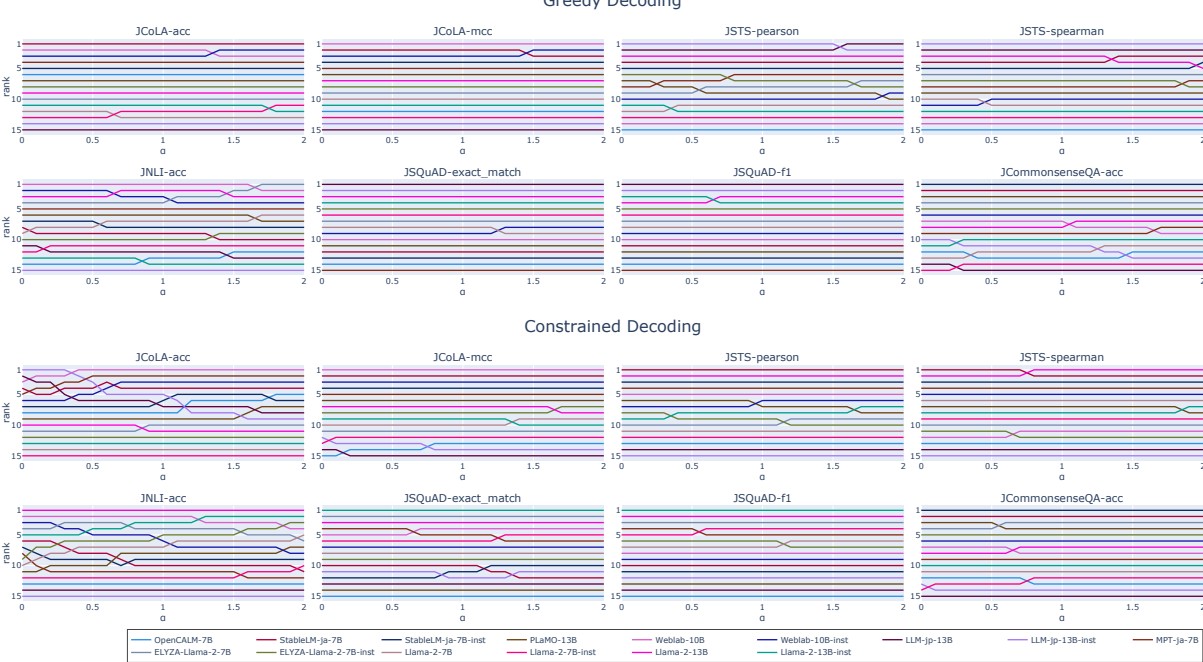

Figure 4: Changes in the rankings of each model when the Sharpe score parameter $\alpha$ is varied from 0 to 2 in increments of 0.1 in the fine-tuning setting on the Japanese dataset. The vertical axis represents the ranking of each model, and the horizontal axis represents $\alpha$. The more intersections of the lines, the greater the variance among the templates. This suggests that the rankings of the models frequently change with the variation of the parameter.

of variance, resulting in the final score being underestimated. Moreover, when there are fluctuations in the rankings between models, a model that has moved down in rank might perform well in the overall score but exhibit large variations in scores for each template. This indicates that a model that has moved up in rank can produce more stable outputs. In Figure 4, we observe that the rankings of the models in JSQuAD and JCommonsenseQA show little change when the parameter $\alpha$ is varied. However, for other datasets such as JNLI, the rankings frequently change with the variation of $\alpha$, indicating a larger variance in evaluation scores among the templates. These results suggest that, while there is generally a correlation between low variance in evaluation scores among templates and high performance when considering only the average of instruction templates, the trend of improvement in performance and variance does not necessarily align for all tasks. Therefore, it was found that the Sharpe score, which considers variance, is an effective performance evaluation metric.

## 6 Conclusion

In this paper, we focused on the variance in the evaluation results of LLMs caused by the variations in instruction templates. We proposed a cross-lingual benchmark dataset based on multiple instruction templates, and reported the evaluation results of models trained on varied data. We also proposed the Sharpe score, which considers the variance in evaluation scores among templates, and demonstrated that it is necessary to consider variance when evaluating LLM performance.

Based on a comparison of diverse LLMs using our dataset and an analysis of the results, we focused on the tasks where cross-lingual knowledge is effective and the effectiveness of LLMs created for specific languages such as Japanese. An issue closely related to what we touched upon in Section 4.1, i.e., the catastrophic forgetting due to continuous training and instruction tuning, is already being studied (Wang et al., 2023; Luo et al., 2023; Kotha et al., 2024), and our dataset may help in analyzing the knowledge and cross-lingual capability of LLMs in more detail. Future LLM development would also benefit from a study verifying the extent of knowledge acquisition and the effects of instruction tuning after different sequences of pre-training, instruction tuning, and continuous training. As a future work, we intend to conduct further analyses and to create a comprehensive evaluation framework for analyzing the NLU capabilities of LLMs by expanding the proposed dataset.

## 7 Limitations

**Coverage of tasks, templates, and languages** This study covered a limited number of tasks, templates, and languages. We conducted a comprehensive validation to demonstrate that evaluation results diverge depending on the variations in instruction templates, highlighting the necessity of evaluations using multiple templates. For the instruction templates used in the evaluation, we utilized the prompt templates from the FLAN dataset, modifying them to create the English evaluation templates and then translating those into the Japanese evaluation templates. In terms of tasks, our study is comprehensive as it covers all the currently accessible tasks in JGLUE, the Japanese standard NLU benchmark dataset, as well as data from comparable English tasks. Although increasing the number of tasks and languages is a direction for future research, obtaining completely aligned data is challenging. Therefore, creating such aligned multilingual datasets and developing evaluation prompt templates for other tasks to increase the number of corresponding tasks will also be future challenges. Moreover, the evaluation prompts were manually created from the FLAN templates. However, a future direction could involve automatically generating evaluation prompts using LLMs such as GPT-4 (OpenAI et al., 2024), phi (Abdin et al., 2024) or Gemini (Team et al., 2023), potentially expanding the range of applicable tasks.

**Number of LLMs used for evaluation** In this study, we evaluated a total of 15 types of LLMs, categorized into four types of language models. Due to the rapid development of LLMs, the number of models continues to increase dramatically, making it impractical to include all results in this study. Therefore, we focused our evaluation on selected language models that cover various training procedures and training data. As discussed in Section 4, we conducted a comprehensive investigation into factors such as transfer performance, the impact of instruction-tuning, continuous training for each language, and the number of parameters. Furthermore, the Sharpe score revealed that the stability of outputs varied across models when considering the variance. Consequently, we believe that the number and quality of language models used in this study are sufficient to demonstrate the necessity of considering output stability in the evaluation of LLMs. To accommodate various future language models, one of the directions we are considering is to create leaderboards and other tools.

**Evaluation of LLMs trained on FLAN templates** Zero-shot evaluation of language models trained on similar data, such as FLAN-T5 (Chung et al., 2024) and FLACUNA (Ghosal et al., 2023), would lead to unfair evaluations as discussed in Section 4.1. Therefore, it would not be appropriate to evaluate such models trained on FLAN data using the evaluation instruction templates created in this study. In contrast, in the fine-tuning setting we used, it is possible to conduct a fair evaluation without considering the effects of pre-training or instruction-tuning data sources, assuming there was no leakage of test data. While we recommend evaluation after fine-tuning, this approach incurs a high computational cost, and therefore developing a mechanism to evaluate zero-shot performance in such models is also desirable and remains a future challenge due to the higher cost of fine-tuning compared to inference.

**Other evaluation paradigms** The performance of LLMs is broadly evaluated along two axes: human-likeness and NLU capabilities. Zheng et al. (2023); Chiang and Lee (2023); Li et al. (2023); Wang et al. (2024) proposed methods that involve evaluating texts generated by LLMs using other LLMs, such as GPT-4 (OpenAI et al., 2024). These evaluation methods focus on human-like dialogue capabilities, emphasizing the models' ability to follow given instructions. Although this study focused solely on NLU capabilities, the stability of outputs is also important for human-like dialogue abilities. We believe that the analysis methods used in this study can be applied to these new evaluation paradigms as well.

## 8 Ethical Considerations

Our evaluation templates are based on the FLAN templates, which are released under the Apache License 2.0, allowing modification and redistribution. We have made modifications, including translations, to these templates. While the original templates were created by the authors of FLAN, we have adapted and extended them for our purposes. The extended templates will be released under the same Apache License 2.0. Moreover, we will only be distributing our modified templates and will not distribute any datasets such as JGLUE, ensuring that there are no licensing issues.

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

## A  Detailed Explanation of Each Task

As shown in Figure 5, we employ five Japanese NLU tasks included in JGLUE (Kurihara et al., 2022)[13] and the corresponding English tasks to evaluate cross-lingual transfer capability and performance of multilingual LLMs: (1) JCoLA (Someya et al., 2024) and CoLA (Warstadt et al., 2019) are linguistic acceptability tasks, where the given sentences are assigned binary labels based on whether they are linguistically acceptable or not. (2) JSTS (Kurihara et al., 2022) and STS-B (Cer et al., 2017) are tasks of judging semantic textual similarity, where similarity scores are assigned to pairs of sentences. (3) JNLI (Kurihara et al., 2022) and MNLI (Williams et al., 2018) are natural language inference tasks, where pairs of sentences are classified as having one of three relationships: entailment, contradiction, or neutrality. (4) JSQuAD (Kurihara et al., 2022) and SQuAD (Rajpurkar et al., 2016) are reading comprehension tasks that require extracting the answer to a question from a given paragraph. (5) JCommonsenseQA (Kurihara et al., 2022) and CommonsenseQA (Talmor et al., 2019) are commonsense reasoning tasks, where the most plausible answer to a question is selected from a set of options. JGLUE was created from scratch based on the methodology used for the corresponding English datasets, ensuring dataset alignment.

## B  Detailed Experimental Settings

**Hyper-parameters**  Table 7 shows the experimental settings of the parameters. We use QLoRA (Dettmers et al., 2023) for fine-tuning. The performance differences between QLoRA and full fine-tuning are minimal (Dettmers et al., 2023; Liu et al., 2024; Dettmers and Zettlemoyer, 2023). Furthermore, we consider QLoRA sufficient for our purpose of evaluating and comparing the LLMs under the same conditions.

**Post-processing**  The post-processing methods and evaluation methods for each task are as follows:

**JCoLA, CoLA**  Parse the generated text according to each regular expression. If this is impossible, assign the label corresponding to "acceptable". The evaluation metrics are accuracy (Acc) and the Matthews correlation coefficient

---

[13]We excluded MARC (Keung et al., 2020) because it is currently unavailable.

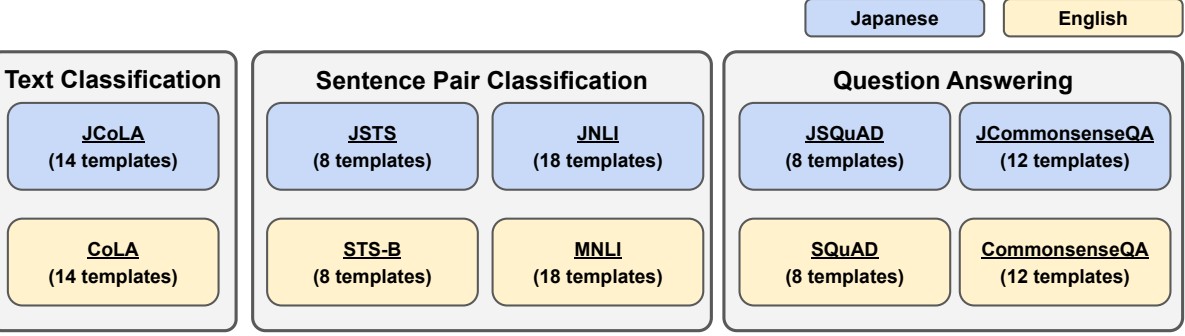

Figure 5: Dataset sources and number of each instruction template.

| Hyper Parameter | Value |
|---|---|
| quant_method | BITS_AND_BYTES |
| load_in_4bit | True |
| bnb_4bit_use_double_quant | True |
| bnb_4bit_quant_type | nf4 |
| bnb_4bit_compute_dtype | float16 |
| lora_alpha | 16 |
| lora_dropout | 0.1 |
| bottleneck_r | 64 |
| optimizer | paged_adamw_8bit |
| batch size | 8 |
| epoch | 1 |
| torch_dtype | float16 |
| lr_scheduler_type | Linear |
| learning_rate | 5e-5 |
| seed | 42 |

Table 7: Hyperparameters used in the experiments. Other parameters were set to their default values. We used the Transformers (Wolf et al., 2020), peft (Mangrulkar et al., 2022), and bitsandbytes (Dettmers et al., 2022) libraries.

(MCC). The score range of accuracy is 0 to 1, while the range of MCC is -1 to 1.

**JSTS, STS-B** Extract parts of the generated text that can be parsed as floats according to the regular expression. If this is impossible, assign a value of 2.0. The evaluation metrics are the Pearson and Spearman correlation coefficients. Both scores range from -1 to 1.

**JNLI, MNLI** Parse the generated text according to each regular expression. If this is impossible, assign the label corresponding to "entailment". The evaluation metric is accuracy.

**JSQuAD, SQuAD** As a general rule, use the original generated text, but if any quotation marks or punctuation are present at the beginning or end of the output text, remove them. Normalize the text to Unicode NFKC. The evaluation metrics are exact match (EM) rate and

F1 score. Both scores range from 0 to 1.

**JCommonsenseQA, CommonsenseQA** Parse the generated text according to the appropriate regular expression. If this is impossible, assign the first of the labels. The evaluation metric is accuracy.

## C  Experimental Results Using Sharpe Score

**Results**  Table 8 and 9 show the results considering the variance among templates using the Sharpe score for the fine-tuning setting on Japanese and English datasets, respectively. Note that $\alpha$ is set to 1, and the corresponding raw results in the same settings are shown in Tables 4 and 6, respectively. Compared to the raw results in Table 4, the evaluation results adjusted by the Sharpe score in Table 8 result in changes in the model ranking. For example, in JNLI with greedy decoding, ELYZA-Llama-2-7B achieves the best evaluation result after adjusting by the Sharpe score in Table 8. Similar change in the model rankings occurs in other cases as well when we use the Sharpe score to consider the variance among instruction templates.

**Ranking on the English dataset**  Figure 6 shows the changes in the rankings among the models, considering the variance among instruction templates, as the Sharpe score parameter $\alpha$ is incremented from 0 to 2 by steps of 0.1 in the English dataset.

In Figure 4, we observe that the rankings of the models in JSQuAD and JCommonsenseQA show little change when the parameter $\alpha$ is varied. However, for other datasets such as JNLI, the rankings frequently change with the variation of $\alpha$, indicating a larger variance in evaluation scores among the templates. This tendency is also observed in the English dataset shown in Figure 6. Specifically,

| | JCoLA Acc/MCC | | JSTS Pearson/Spearman | | JNLI Acc | | JSQuAD EM/F1 | | JCommonsenseQA Acc | |
| --- | --- | --- | --- | --- | --- | --- | --- | --- | --- | --- |
| Model | Greedy | Constrained | Greedy | Constrained | Greedy | Constrained | Greedy | Constrained | Greedy | Constrained |
| OpenCALM-7B | 0.833/0.251 | 0.838/0.202 | 0.898/0.858 | 0.820/0.769 | 0.869 | 0.854 | 0.814/0.905 | 0.794/0.900 | 0.838 | 0.825 |
| StableLM-ja-7B | **0.850**/0.421 | 0.845/0.418 | 0.920/0.887 | **0.901**/0.877 | 0.901 | 0.901 | 0.873/0.948 | 0.865/0.941 | 0.922 | 0.921 |
| StableLM-ja-7B-inst | 0.841/0.398 | 0.837/0.398 | 0.919/0.885 | 0.900/0.875 | 0.903 | 0.903 | 0.870/0.946 | 0.867/0.940 | **0.923** | **0.925** |
| PLaMo-13B | 0.831/0.356 | 0.832/0.351 | 0.914/0.878 | 0.892/0.864 | 0.904 | 0.905 | 0.877/0.947 | 0.848/0.936 | 0.906 | 0.903 |
| Weblab-10B | 0.848/**0.445** | **0.848/0.444** | 0.906/0.866 | 0.893/0.853 | 0.909 | 0.909 | 0.884/0.951 | 0.881/0.946 | 0.885 | 0.887 |
| Weblab-10B-inst | 0.849/0.423 | 0.847/0.416 | 0.914/0.875 | 0.893/0.866 | 0.908 | 0.905 | 0.887/0.953 | 0.878/0.946 | 0.893 | 0.891 |
| LLM-jp-13B | 0.302/0.161 | 0.828/0.152 | **0.929**/0.897 | 0.612/0.519 | 0.862 | 0.317 | **0.907/0.963** | 0.857/0.936 | 0.742 | 0.349 |
| LLM-jp-13B-inst | 0.302/0.162 | 0.825/0.166 | **0.929**/**0.899** | -0.103/-0.042 | 0.810 | 0.300 | 0.904/0.962 | 0.866/0.938 | 0.830 | 0.740 |
| MPT-ja-7B | 0.844/0.384 | 0.847/0.379 | 0.917/0.882 | 0.898/0.871 | 0.906 | 0.900 | 0.002/0.463 | 0.880/0.948 | 0.886 | 0.886 |
| ELYZA-Llama-2-7B | 0.818/0.292 | 0.820/0.311 | 0.916/0.884 | 0.890/0.854 | **0.910** | 0.909 | 0.888/0.954 | **0.888**/0.952 | 0.898 | 0.910 |
| ELYZA-Llama-2-7B-inst | 0.825/0.318 | 0.814/0.330 | 0.916/0.882 | 0.889/0.850 | 0.902 | 0.910 | 0.893/0.957 | 0.872/0.947 | 0.896 | 0.897 |
| Llama-2-7B | 0.801/0.290 | 0.812/0.318 | 0.911/0.874 | 0.888/0.865 | 0.905 | 0.909 | 0.886/0.954 | 0.875/0.948 | 0.845 | 0.853 |
| Llama-2-7B-inst | 0.804/0.234 | 0.782/0.238 | 0.906/0.870 | 0.885/0.861 | 0.892 | 0.902 | 0.889/0.956 | 0.881/0.949 | 0.821 | 0.838 |
| Llama-2-13B | 0.825/0.335 | 0.817/0.330 | 0.921/0.885 | **0.901/0.878** | 0.909 | **0.917** | 0.899/0.961 | **0.888**/0.953 | 0.887 | 0.888 |
| Llama-2-13B-inst | 0.803/0.276 | 0.813/0.316 | 0.907/0.871 | 0.893/0.865 | 0.862 | 0.912 | 0.894/0.960 | **0.888/0.954** | 0.865 | 0.881 |

Table 8: Adjusted evaluation results using the Sharpe score in the fine-tuning setting on Japanese datasets.

| | CoLA Acc/MCC | | STS-B Pearson/Spearman | | MNLI Acc | | SQuAD EM/F1 | | CommonsenseQA Acc | |
| --- | --- | --- | --- | --- | --- | --- | --- | --- | --- | --- |
| Model | Greedy | Constrained | Greedy | Constrained | Greedy | Constrained | Greedy | Constrained | Greedy | Constrained |
| PLaMO-13B | 0.829/0.605 | 0.833/0.612 | 0.888/0.888 | 0.862/0.863 | 0.790 | 0.786 | 0.753/0.910 | 0.738/0.906 | 0.723 | 0.720 |
| Weblab-10B | 0.832/0.604 | 0.831/0.602 | 0.894/0.896 | 0.882/0.884 | 0.795 | 0.796 | 0.761/0.911 | 0.733/0.902 | 0.650 | 0.649 |
| Weblab-10B-inst | 0.825/0.585 | 0.822/0.579 | 0.903/0.903 | 0.905/0.905 | 0.812 | 0.812 | 0.760/0.914 | 0.733/0.904 | 0.657 | 0.655 |
| LLM-jp-13B | 0.375/0.190 | 0.666/0.190 | 0.890/0.894 | 0.547/0.595 | 0.557 | 0.413 | 0.435/0.788 | 0.757/0.913 | 0.594 | 0.301 |
| LLM-jp-13B-inst | 0.375/0.191 | 0.665/0.193 | **0.909/0.907** | 0.876/0.881 | 0.559 | 0.397 | 0.410/0.779 | 0.752/0.912 | 0.622 | 0.336 |
| MPT-ja-7B | 0.808/0.545 | 0.805/0.538 | 0.898/0.898 | 0.883/0.885 | 0.785 | 0.718 | 0.000/0.492 | 0.731/0.901 | 0.694 | 0.694 |
| ELYZA-Llama-2-7B | 0.835/0.617 | 0.846/0.641 | 0.889/0.891 | 0.909/**0.912** | 0.846 | 0.828 | 0.788/0.928 | 0.769/0.923 | 0.749 | 0.755 |
| ELYZA-Llama-2-7B-inst | 0.841/0.633 | 0.855/**0.669** | 0.885/0.887 | **0.912**/0.911 | 0.849 | 0.851 | 0.788/0.928 | 0.767/0.922 | 0.746 | 0.748 |
| Llama-2-7B | 0.833/0.611 | 0.849/0.651 | 0.884/0.887 | 0.890/0.894 | 0.845 | 0.849 | 0.794/0.931 | 0.772/0.924 | 0.758 | 0.762 |
| Llama-2-7B-inst | 0.841/0.629 | 0.844/0.636 | 0.905/0.905 | 0.887/0.890 | 0.848 | 0.855 | 0.794/0.931 | 0.773/0.923 | 0.750 | 0.758 |
| Llama-2-13B | **0.851/0.653** | **0.857**/0.667 | 0.888/0.890 | 0.901/0.902 | **0.865** | **0.871** | **0.800/0.937** | **0.784**/0.932 | **0.796** | **0.792** |
| Llama-2-13B-inst | 0.836/0.623 | 0.849/0.647 | 0.894/0.894 | 0.896/0.900 | 0.864 | 0.866 | 0.793/0.934 | **0.784/0.932** | 0.769 | 0.788 |

Table 9: Adjusted evaluation results using the Sharpe score in the fine-tuning setting on English datasets.

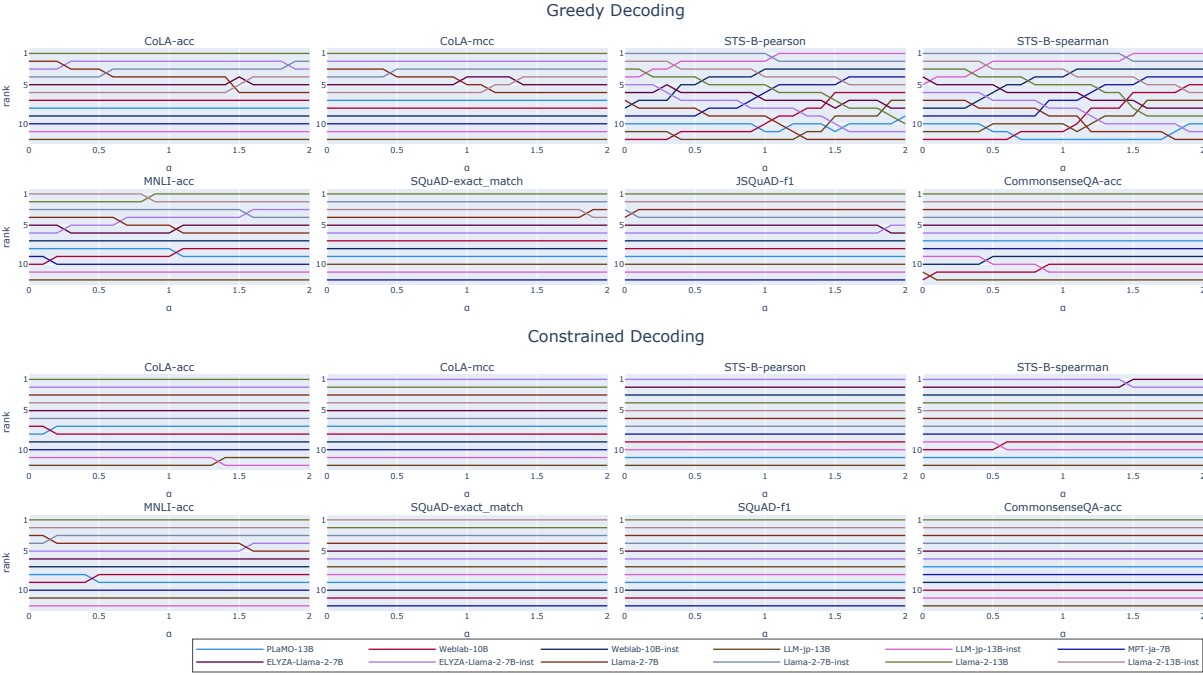

Figure 6: Changes in the rankings of each model when the Sharpe score parameter $\alpha$ is varied from 0 to 2 in increments of 0.1 in the fine-tuning setting on the English dataset.

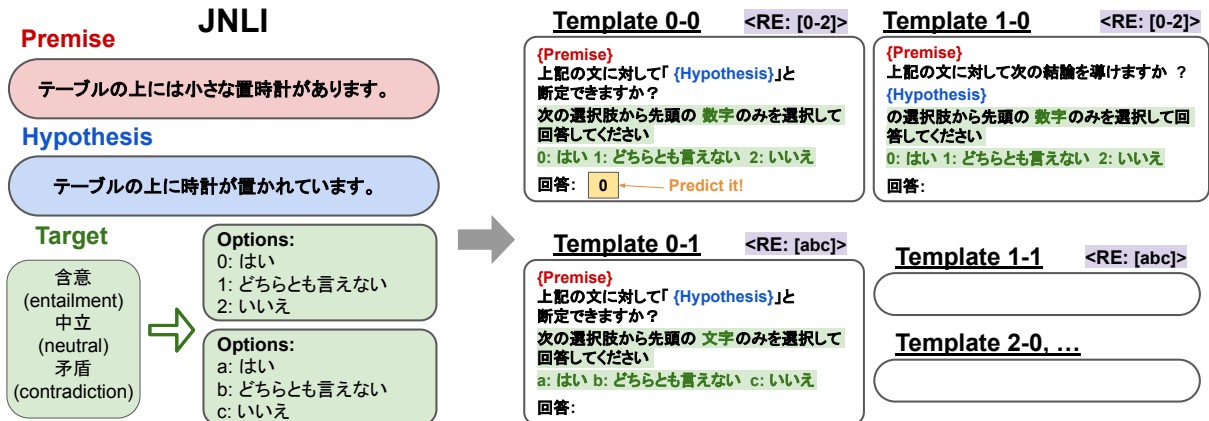

Figure 7: The examples of the dataset creation process for the JNLI task. We manually translated the template of MNLI in Figure 1 to create the template of JNLI.

with greedy decoding, evaluation results have a significant variance due to the types of instruction templates in STS-B and MNLI.

## D Discussions (Details)

### D.1 Model Size

Based on the comparison of the 13B model group with the 7B model group, it cannot be concluded that an increase in parameters necessarily affects NLU performance. However, if we compare models based solely on the number of parameters within the Llama-2 series, the increase in evaluation scores relative to the increase in parameters is minimal. On the other hand, when comparing PLaMO-13B with StableLM-ja-7B, despite the difference in the number of parameters, StableLM-ja-7B achieves higher performance. This suggests that improvements in NLU performance are more significantly influenced by the training data than by the number of parameters. These results are in line with recent studies (Hoffmann et al., 2024; Xue et al., 2023) that indicate that the quantity of training data is more effective than the number of parameters.

### D.2 Language Transfer Capability

When discussing the cross-lingual transfer capability in Sections 4.1 and 4.2, we noted that LLM-jp-13B-inst (results in Table 5), trained with the instruction-tuning dataset Jaster, which is based on JGLUE, can make certain inferences even in the zero-shot setting through cross-lingual transfer, despite not being trained on the corresponding English data for STS-B and CommonsenseQA. For STS-B, the results are comparable to those discussed for Llama2-13B in Section 4.1, demonstrating similar transfer performance from Japanese to

English. For CommonsenseQA, the model could likely make correct inferences because some commonsense knowledge is shared between Japanese and English. This indicates that when NLU tasks are explicitly learned for a specific language, the performance can be transferred to some extent to other languages. It remains a future challenge, however, to identify the domains where cross-lingual transfer is possible.

## E Example of Japanese Instruction Template

Figure 7 shows examples of the dataset creation process for JNLI tasks. We created Japanese JNLI templates by manually translating the MNLI templates corresponding to the English tasks, as shown in Figure 1. For instance, JNLI provides pairs of sentences, a premise, and a hypothesis. We then apply each instruction template to these sentence pairs to create natural language sentences to be used as input sequences. The expected output format for answers follows FLAN. We convert the answer labels to conversational text and instruct the LLMs to generate only the corresponding number or letter.

## F Examples of All Evaluation Templates

The evaluation templates are presented as follows: CoLA and JCoLA in Tables 10 and 11; STS-B and JSTS in Tables 12, and 13; MNLI and JNLI in Tables 14, 15 and 16; SQuAD and JSQuAD in Table 17; and CommonsenseQA and JCommonsenseQA in Tables 18 and 19. The elements inside the curly brackets are replaced with questions, sentences, or contexts provided as minimal information for each task. Please refer to each task for the specific elements.

| ID | CoLA | JCoLA | Constraint |
|---|---|---|---|
| 0-0 | Sentence: "{sentence}"
Would a linguist rate this sentence to be acceptable linguistically?

Answer using only the leading number of one of the following options.
0: Yes, 1: No
Answer: | 文: 「{sentence}」
言語学者がこの文を言語学的に受け入れると思いますか?

次の選択肢から先頭の数字のみを選択して回答してください。
0: はい、1: いいえ
回答: | [0-1] |
| 0-1 | Sentence: "{sentence}"
Would a linguist rate this sentence to be acceptable linguistically?

Answer using only the leading letter of one of the following options.
a: Yes, b: No
Answer: | 文: 「{sentence}」
言語学者がこの文を言語学的に受け入れると思いますか?

次の選択肢から先頭の英字のみを選択して回答してください。
a: はい、b: いいえ
回答: | [ab] |
| 1-0 | {sentence}
How would you consider the linguistic integrity of the preceding sentence?

Answer using only the leading number of one of the following options.
0: Yes, 1: No
Answer: | {sentence}
あなたは前の文に言語学的な整合性があると思いますか?

次の選択肢から先頭の数字のみを選択して回答してください。
0: はい、1: いいえ
回答: | [0-1] |
| 1-1 | {sentence}
How would you consider the linguistic integrity of the preceding sentence?

Answer using only the leading letter of one of the following options.
a: Yes, b: No
Answer: | {sentence}
あなたは前の文に言語学的な整合性があると思いますか?

次の選択肢から先頭の英字のみを選択して回答してください。
a: はい、b: いいえ
回答: | [ab] |
| 2-0 | Test sentence: "{sentence}"
Is this test sentence a correct grammatical English sentence?

Answer using only the leading number of one of the following options.
0: Yes, 1: No
Answer: | テスト文: 「{sentence}」
このテスト文は日本語の文法を満たす正しい文ですか?

次の選択肢から先頭の数字のみを選択して回答してください。
0: はい、1: いいえ
回答: | [0-1] |
| 2-1 | Test sentence: "{sentence}"
Is this test sentence a correct grammatical English sentence?

Answer using only the leading letter of one of the following options.
a: Yes, b: No
Answer: | テスト文: 「{sentence}」
このテスト文は日本語の文法を満たす正しい文ですか?

次の選択肢から先頭の英字のみを選択して回答してください。
a: はい、b: いいえ
回答: | [ab] |
| 3-0 | Is the following sentence linguistically acceptable?
{sentence}

Answer using only the leading number of one of the following options.
0: Yes, 1: No
Answer: | 次の文は言語学的に受け入れられますか?
{sentence}

次の選択肢から先頭の数字のみを選択して回答してください。
0: はい、1: いいえ
回答: | [0-1] |

Table 10: The evaluation templates for CoLA and JCoLA (Part 1 of 2).

| ID | CoLA | JCoLA | Constraint |
|---|---|---|---|
| 3-1 | Is the following sentence linguistically acceptable? {sentence}

Answer using only the leading letter of one of the following options.
a: Yes, b: No
Answer: | 次の文は言語学的に受け入れられますか? {sentence}

次の選択肢から先頭の英字のみを選択して回答してください。
a: はい、b: いいえ
回答: | [ab] |
| 4-0 | Would the following sentence, by the strictest standards, be considered correct by a linguist?

{sentence}
Answer using only the leading number of one of the following options.
0: Yes, 1: No
Answer: | 厳密な基準において言語学者は以下の文を正しいと判断すると思いますか?

{sentence}
次の選択肢から先頭の数字のみを選択して回答してください。
0: はい、1: いいえ
回答: | [0-1] |
| 4-1 | Would the following sentence, by the strictest standards, be considered correct by a linguist?

{sentence}
Answer using only the leading letter of one of the following options.
a: Yes, b: No
Answer: | 厳密な基準において言語学者は以下の文を正しいと判断すると思いますか?

{sentence}
次の選択肢から先頭の英字のみを選択して回答してください。
a: はい、b: いいえ
回答: | [ab] |
| 5-0 | Is the next sentence syntactically and semantically acceptable?

{sentence}
Answer using only the leading number of one of the following options.
0: Yes, 1: No
Answer: | 厳密な基準において言語学者は以下の文を正しいと判断すると思いますか?

{sentence}
次の選択肢から先頭の数字のみを選択して回答してください。
0: はい、1: いいえ
回答: | [0-1] |
| 5-1 | Is the next sentence syntactically and semantically acceptable?

{sentence}
Answer using only the leading letter of one of the following options.
a: Yes, b: No
Answer: | 次の文は統語的にも意味的にも受け入れることができますか?

{sentence}
次の選択肢から先頭の英字のみを選択して回答してください。
a: はい、b: いいえ
回答: | [ab] |
| 6-0 | Would a linguist find the following sentence to be a valid English sentence grammatically?

{sentence}
Answer using only the leading number of one of the following options.
0: Yes, 1: No
Answer: | 言語学者は以下の文を文法的に妥当な日本語の文として認めると思いますか?

{sentence}
次の選択肢から先頭の数字のみを選択して回答してください。
0: はい、1: いいえ
回答: | [0-1] |
| 6-1 | Would a linguist find the following sentence to be a valid English sentence grammatically?

{sentence}
Answer using only the leading letter of one of the following options.
a: Yes, b: No
Answer: | 言語学者は以下の文を文法的に妥当な日本語の文として認めると思いますか?

{sentence}
次の選択肢から先頭の英字のみを選択して回答してください。
a: はい、b: いいえ
回答: | [ab] |

Table 11: The evaluation templates for CoLA and JCoLA (Part 2 of 2).

| ID | STS-B | JSTS | Constraint |
|---|---|---|---|
| 0-0 | {sentence1}
{sentence2}

Rate the textual similarity of these two sentences on a scale from 0 to 5, where 0 is "no meaning overlap" and 5 is "means the same thing".

Answer on a scale from 0.000 to 5.000 with 0.001 increments.
Answer: | {sentence1}
{sentence2}

この2つの文の類似度を0.0から5.0までのスコアで評価してください。なお、0.0を「意味が重複していない」、5.0を「同じ意味である」とします。

0.0から5.0までのスコアを0.1刻みで回答してください。
回答: | $([0-4]\backslash.[0-9]\{3\}\|5.0)$ |
| 1-0 | {sentence1}
{sentence2}

On a scale from 0 to 5, where 0 is "no meaning overlap" and 5 is "means the same thing", how closely does the first sentence resemble the second one?

Answer on a scale from 0.000 to 5.000 with 0.001 increments.
Answer: | {sentence1}
{sentence2}

0.0から5.0までのスコアで0.0を「意味が重複していない」、5.0を「同じ意味である」としたとき、最初の文は二つ目の文にどれだけ似ていますか？

0.0から5.0までのスコアを0.1刻みで回答してください。
回答: | $([0-4]\backslash.[0-9]\{3\}\|5.0)$ |
| 2-0 | Sentence 1: {sentence1}
Sentence 2: {sentence2}

From 0 to 5 (0="no meaning overlap" and 5="means the same thing"), how similar are the two sentences?

Answer on a scale from 0.000 to 5.000 with 0.001 increments.
Answer: | 文1: {sentence1}
文2: {sentence2}

0.0から5.0までのスコアによる評価(0.0=意味が重複しない、5.0=同じ意味である)において、この二つの文はどれだけ似ていますか？

0.0から5.0までのスコアを0.1刻みで回答してください。
回答: | $([0-4]\backslash.[0-9]\{3\}\|5.0)$ |
| 3-0 | How similar are the following two sentences?

{sentence1}
{sentence2}

Give the answer on a scale from 0 - 5, where 0 is "not similar at all" and 5 is "means the same thing".

Answer on a scale from 0.000 to 5.000 with 0.001 increments.
Answer: | 次の二つの文はどれだけ似ていますか？

{sentence1}
{sentence2}

0.0から5.0までのスコアで評価してください。0.0は「全く似ていない」、5.0は「同じ意味である」をそれぞれ表しています。

0.0から5.0までのスコアを0.1刻みで回答してください。
回答: | $([0-4]\backslash.[0-9]\{3\}\|5.0)$ |

Table 12: The evaluation templates for STS-B and JSTS (Part 1 of 2).

| ID | STS-B | JSTS | Constraint |
|---|---|---|---|
| 4-0 | Do the following sentences say the same thing?

{sentence1}
{sentence2}

Return your answer on a scale from 0 to 5, where 0 is "not similar" and 5 is "very similar".

Answer on a scale from 0.000 to 5.000 with 0.001 increments.
Answer: | 次の二つの文は同じ内容を表していますか？

{sentence1}
{sentence2}

あなたの回答を0.0から5.0までのスコアで評価してください。0.0は「全く似ていない」、5.0は「とても似ている」をそれぞれ表しています。

0.0から5.0までのスコアを0.1刻みで回答してください。
回答: | $([0-4]\backslash.[0-9]\{3\}\|5.0)$ |
| 5-0 | Rate the similarity of the following two sentences on a scale from 0 to 5, where 0 is "no meaning overlap" and 5 is "means the same thing"?

{sentence1}
{sentence2}

Answer on a scale from 0.000 to 5.000 with 0.001 increments.
Answer: | 次の二つの文の類似度を0.0から5.0までのスコアで評価してください。0.0は「意味に被りがない」、5.0は「同じ意味を表している」をそれぞれ表しています。

{sentence1}
{sentence2}

0.0から5.0までのスコアを0.1刻みで回答してください。
回答: | $([0-4]\backslash.[0-9]\{3\}\|5.0)$ |
| 6-0 | On a scale from 0-5, where 0 is "not similar" and 5 is "very similar", how similar is the sentence "{sentence1}" to the sentence "{sentence2}"?

Answer on a scale from 0.000 to 5.000 with 0.001 increments.
Answer: | 0.0から5.0までのスコアで、0.0を「似ていない」、5.0を「似ている」とするとき、文「{sentence1}」と文「{sentence2}」はどれだけ似ていますか?

0.0から5.0までのスコアを0.1刻みで回答してください。
回答: | $([0-4]\backslash.[0-9]\{3\}\|5.0)$ |
| 7-0 | How similar are these two sentences, on a scale from 0-5 (0 is "not similar" and 5 is "very similar")?

{sentence1}
{sentence2}

Answer on a scale from 0.000 to 5.000 with 0.001 increments.
Answer: | 次の二つの文は0.0から5.0までのスコア（0.0は「似ていない」、5.0は「非常に似ている」）で、どれだけ似ていますか？

{sentence1}
{sentence2}

0.0から5.0までのスコアを0.1刻みで回答してください。
回答: | $([0-4]\backslash.[0-9]\{3\}\|5.0)$ |

Table 13: The evaluation templates for STS-B and JSTS (Part 2 of 2).

| ID | MNLI | JNLI | Constraint |
|---|---|---|---|
| 0-0 | {sentence1}

Based on the sentence above can we conclude that "{sentence2}"?

Answer using only the leading number of one of the following options.
0: Yes, 1: It's impossible to say, 2: No
Answer: | {sentence1}

上記の文に対して「{sentence2}」と断定できますか?

次の選択肢から先頭の数字のみを選択して回答してください。
0: はい、1: どちらとも言えない、2: いいえ
回答: | [0-2] |
| 0-1 | {sentence1}

Based on the sentence above can we conclude that "{sentence2}"?

Answer using only the leading letter of one of the following options.
a: Yes, b: It's impossible to say, c: No
Answer: | {sentence1}

上記の文に対して「{sentence2}」と断定できますか?

次の選択肢から先頭の英字のみを選択して回答してください。
a: はい、b: どちらとも言えない、c: いいえ
回答: | [abc] |
| 1-0 | {sentence1}

Based on that sentence can we conclude that this sentence is true?
{sentence2}

Answer using only the leading number of one of the following options.
0: Yes, 1: It's impossible to say, 2: No
Answer: | {sentence1}

上記の文に対して次の文が真実であると断定できますか?
{sentence2}

次の選択肢から先頭の数字のみを選択して回答してください。
0: はい、1: どちらとも言えない、2: いいえ
回答: | [0-2] |
| 1-1 | {sentence1}

Based on that sentence can we conclude that this sentence is true?
{sentence2}

Answer using only the leading letter of one of the following options.
a: Yes, b: It's impossible to say, c: No
Answer: | {sentence1}

上記の文に対して次の文が真実であると断定できますか?
{sentence2}

次の選択肢から先頭の英字のみを選択して回答してください。
a: はい、b: どちらとも言えない、c: いいえ
回答: | [abc] |
| 2-0 | {sentence1}

Can we draw the following conclusion?
{sentence2}

Answer using only the leading number of one of the following options.
0: Yes, 1: It's impossible to say, 2: No
Answer: | {sentence1}

上記の文に対して次の結論を導けますか?
{sentence2}

次の選択肢から先頭の数字のみを選択して回答してください。
0: はい、1: どちらとも言えない、2: いいえ
回答: | [0-2] |
| 2-1 | {sentence1}

Can we draw the following conclusion?
{sentence2}

Answer using only the leading letter of one of the following options.
a: Yes, b: It's impossible to say, c: No
Answer: | {sentence1}

上記の文に対して次の結論を導けますか?
{sentence2}

次の選択肢から先頭の英字のみを選択して回答してください。
a: はい、b: どちらとも言えない、c: いいえ
回答: | [abc] |

Table 14: The evaluation templates for MNLI and JNLI (Part 1 of 3).

| ID | MNLI | JNLI | Constraint |
|---|---|---|---|
| 3-0 | {sentence1}

Does this next sentence follow, given the preceding text?
{sentence2}

Answer using only the leading number of one of the following options.
0: Yes, 1: It's impossible to say, 2: No
Answer: | {sentence1}

次の文は上記の文に沿っていますか?
{sentence2}

次の選択肢から先頭の数字のみを選択して回答してください。
0: はい、1: どちらとも言えない、2: いいえ
回答: | [0-2] |
| 3-1 | {sentence1}

Does this next sentence follow, given the preceding text?
{sentence2}

Answer using only the leading letter of one of the following options.
a: Yes, b: It's impossible to say, c: No
Answer: | {sentence1}

次の文は上記の文に沿っていますか?
{sentence2}

次の選択肢から先頭の英字のみを選択して回答してください。
a: はい、b: どちらとも言えない、c: いいえ
回答: | [abc] |
| 4-0 | {sentence1}

Can we infer the following?
{sentence2}

Answer using only the leading number of one of the following options.
0: Yes, 1: It's impossible to say, 2: No
Answer: | {sentence1}

上記の文から次の文を導けますか?
{sentence2}

次の選択肢から先頭の数字のみを選択して回答してください。
0: はい、1: どちらとも言えない、2: いいえ
回答: | [0-2] |
| 4-1 | {sentence1}

Can we infer the following?
{sentence2}

Answer using only the leading letter of one of the following options.
a: Yes, b: It's impossible to say, c: No
Answer: | {sentence1}

上記の文から次の文を導けますか?
{sentence2}

次の選択肢から先頭の英字のみを選択して回答してください。
a: はい、b: どちらとも言えない、c: いいえ
回答: | [abc] |
| 5-0 | Read the following sentence and determine if the hypothesis is true:

{sentence1}

Hypothesis: {sentence2}

Answer using only the leading number of one of the following options.
0: Yes, 1: It's impossible to say, 2: No
Answer: | 次の文を読んで仮説が正しいか判断してください:

{sentence1}

仮説: {sentence2}

次の選択肢から先頭の数字のみを選択して回答してください。
0: はい、1: どちらとも言えない、2: いいえ
回答: | [0-2] |
| 5-1 | Read the following sentence and determine if the hypothesis is true:

{sentence1}

Hypothesis: {sentence2}

Answer using only the leading letter of one of the following options.
a: Yes, b: It's impossible to say, c: No
Answer: | 次の文を読んで仮説が正しいか判断してください:

{sentence1}

仮説: {sentence2}

次の選択肢から先頭の英字のみを選択して回答してください。
a: はい、b: どちらとも言えない、c: いいえ
回答: | [abc] |

Table 15: The evaluation templates for MNLI and JNLI (Part 2 of 3).

| ID | MNLI | JNLI | Constraint |
|---|---|---|---|
| 6-0 | Read the text and determine if the sentence is true:

{sentence1}

Sentence: {sentence2}

Answer using only the leading number of the following options.
0: Yes, 1: It's impossible to say, 2: No
Answer: | 次の文を読んで与えられた文が正しいか判断してください:

{sentence1}

文: {sentence2}

次の選択肢から先頭の数字のみを選択して回答してください。
0: はい、1: どちらとも言えない、2: いいえ
回答: | [0-2] |
| 6-1 | Read the text and determine if the sentence is true:

{sentence1}

Sentence: {sentence2}

Answer using only the leading letter of one of the following options.
a: Yes, b: It's impossible to say, c: No
Answer: | 次の文を読んで与えられた文が正しいか判断してください:

{sentence1}

文: {sentence2}

次の選択肢から先頭の英字のみを選択して回答してください。
a: はい、b: どちらとも言えない、c: いいえ
回答: | [abc] |
| 7-0 | Can we draw the following hypothesis from the context?

Context: {sentence1}

Hypothesis: {sentence2}

Answer using only the leading number of one of the following options.
0: Yes, 1: It's impossible to say, 2: No
Answer: | 与えられた文脈から後続する仮説を導けますか?

文脈: {sentence1}

仮説: {sentence2}

次の選択肢から先頭の数字のみを選択して回答してください。
0: はい、1: どちらとも言えない、2: いいえ
回答: | [0-2] |
| 7-1 | Can we draw the following hypothesis from the context?

Context: {sentence1}

Hypothesis: {sentence2}

Answer using only the leading letter of one of the following options.
a: Yes, b: It's impossible to say, c: No
Answer: | 与えられた文脈から後続する仮説を導けますか?

文脈: {sentence1}

仮説: {sentence2}

次の選択肢から先頭の英字のみを選択して回答してください。
a: はい、b: どちらとも言えない、c: いいえ
回答: | [abc] |
| 8-0 | Determine if the sentence is true based on the text below:
{sentence2}

{sentence1}
Answer using only the leading number of one of the following options.
0: Yes, 1: It's impossible to say, 2: No
Answer: | 以下の文から、この文が正しいか判断してください。:
{sentence2}

{sentence1}
次の選択肢から先頭の数字のみを選択して回答してください。
0: はい、1: どちらとも言えない、2: いいえ
回答: | [0-2] |
| 8-1 | Determine if the sentence is true based on the text below:
{sentence2}

{sentence1}
Answer using only the leading letter of one of the following options.
a: Yes, b: It's impossible to say, c: No
Answer: | 以下の文から、この文が正しいか判断してください。:
{sentence2}

{sentence1}
次の選択肢から先頭の英字のみを選択して回答してください。
a: はい、b: どちらとも言えない、c: いいえ
回答: | [abc] |

Table 16: The evaluation templates for MNLI and JNLI (Part 3 of 3).

| ID | SQuAD | JSQuAD | Constraint |
|---|---|---|---|
| 0-0 | Please answer a question about the following article about "{title}":

{context}

{question}
Extract the answer from the text above.
Answer: | 以下の「{title}」に関する記事について次の質問に回答してください。

記事: {context}

質問: {question}
上記のテキストから抜き出して回答してください。
回答: | .+ |
| 1-0 | Read this and answer the question

{context}

{question}

Extract the answer from the text above.
Answer: | 次を読み質問に答えてください。

{context}

質問: {question}

上記のテキストから抜き出して回答してください。
回答: | .+ |
| 2-0 | {context}

{question}

Extract the answer from the text above.
Answer: | {context}

{question}

上記のテキストから抜き出して回答してください。
回答: | .+ |
| 3-0 | Answer a question about this article:
{context}

{question}

Extract the answer from the text above.
Answer: | この記事に関する質問に答えてください:
{context}

{question}

上記のテキストから抜き出して回答してください。
回答: | .+ |
| 4-0 | Here is a question about this article: {context}
What is the answer to this question: {question}

Extract the answer from the text above.
Answer: | この記事についての質問です: {context}
この質問に対する答えは何ですか: {question}

上記のテキストから抜き出して回答してください。
回答: | .+ |
| 5-0 | Article: {context}

Question: {question}

Extract the answer from the text above.
Answer: | 記事: {context}

質問: {question}

上記のテキストから抜き出して回答してください。
回答: | .+ |
| 6-0 | Article: {context}

Now answer this question: {question}

Extract the answer from the text above.
Answer: | 記事: {context}

では次の質問に答えてください: {question}

上記のテキストから抜き出して回答してください。
回答: | .+ |
| 7-0 | {title}
{context}

Q: {question}

Extract the answer from the text above.
Answer: | {title}
{context}

質問: {question}

上記のテキストから抜き出して回答してください。
回答: | .+ |

Table 17: The evaluation templates for SQuAD and JSQuAD.

| ID | CommonsenseQA | JCommonsenseQA | Constraint |
|---|---|---|---|
| 0-0 | {question}

Answer using only the leading number of one of the following options.
0: {choice0}, 1: {choice1}, 2: {choice2}, 3: {choice3}, 4: {choice4}
Answer: | {question}

次の選択肢から先頭の数字のみを選択して回答してください。
0: {choice0}、1: {choice1}、2: {choice2}、3: {choice3}、4: {choice4}
回答: | [0-4] |
| 0-1 | {question}

Answer using only the leading letter of one of the following options.
a: {choice0}, b: {choice1}, c: {choice2}, d: {choice3}, e: {choice4}
Answer: | {question}

次の選択肢から先頭の英字のみを選択して回答してください。
a: {choice0}、b: {choice1}、c: {choice2}、d: {choice3}、e: {choice4}
回答: | [abcde] |
| 1-0 | Question: {question}

Answer using only the leading number of one of the following options.
0: {choice0}, 1: {choice1}, 2: {choice2}, 3: {choice3}, 4: {choice4}
Answer: | 質問: {question}

次の選択肢から先頭の数字のみを選択して回答してください。
0: {choice0}、1: {choice1}、2: {choice2}、3: {choice3}、4: {choice4}
回答: | [0-4] |
| 1-1 | Question: {question}

Answer using only the leading letter of one of the following options.
a: {choice0}, b: {choice1}, c: {choice2}, d: {choice3}, e: {choice4}
Answer: | 質問: {question}

次の選択肢から先頭の英字のみを選択して回答してください。
a: {choice0}、b: {choice1}、c: {choice2}、d: {choice3}、e: {choice4}
回答: | [abcde] |
| 2-0 | Question: {question}

What is the correct answer to the question from the following choices?
Answer using only the leading number of one of the following options.
0: {choice0}, 1: {choice1}, 2: {choice2}, 3: {choice3}, 4: {choice4}
Answer: | 質問: {question}

次の選択肢の中で正しい答えはどれですか?
次の選択肢から先頭の数字のみを選択して回答してください。
0: {choice0}、1: {choice1}、2: {choice2}、3: {choice3}、4: {choice4}
回答: | [0-4] |
| 2-1 | Question: {question}

What is the correct answer to the question from the following choices?
Answer using only the leading letter of one of the following options.
a: {choice0}, b: {choice1}, c: {choice2}, d: {choice3}, e: {choice4}
Answer: | 質問: {question}

次の選択肢の中で正しい答えはどれですか?
次の選択肢から先頭の英字のみを選択して回答してください。
a: {choice0}、b: {choice1}、c: {choice2}、d: {choice3}、e: {choice4}
回答: | [abcde] |

Table 18: The evaluation templates for CommonsenseQA and JCommonsenseQA (Part 1 of 2).

| ID | CommonsenseQA | JCommonsenseQA | Constraint |
|---|---|---|---|
| 3-0 | Q: {question}

What is the correct answer to this question?
Answer using only the leading number of one of the following options.
0: {choice0}, 1: {choice1}, 2: {choice2}, 3: {choice3}, 4: {choice4}
Answer: | 質問: {question}

この質問に対する正しい答えは何ですか?
次の選択肢から先頭の数字のみを選択して回答してください。
0: {choice0}、1: {choice1}、2: {choice2}、3: {choice3}、4: {choice4}
回答: | [0-4] |
| 3-1 | Q: {question}

What is the correct answer to this question?
Answer using only the leading letter of one of the following options.
a: {choice0}, b: {choice1}, c: {choice2}, d: {choice3}, e: {choice4}
Answer: | 質問: {question}

この質問に対する正しい答えは何ですか?
次の選択肢から先頭の英字のみを選択して回答してください。
a: {choice0}、b: {choice1}、c: {choice2}、d: {choice3}、e: {choice4}
回答: | [abcde] |
| 4-0 | What is the answer?

{question}

Answer using only the leading number of one of the following options.
0: {choice0}, 1: {choice1}, 2: {choice2}, 3: {choice3}, 4: {choice4}
Answer: | 何が答えですか?

{question}

次の選択肢から先頭の数字のみを選択して回答してください。
0: {choice0}、1: {choice1}、2: {choice2}、3: {choice3}、4: {choice4}
回答: | [0-4] |
| 4-1 | What is the answer?

{question}

Answer using only the leading letter of one of the following options.
a: {choice0}, b: {choice1}, c: {choice2}, d: {choice3}, e: {choice4}
Answer: | 何が答えですか?

{question}

次の選択肢から先頭の英字のみを選択して回答してください。
a: {choice0}、b: {choice1}、c: {choice2}、d: {choice3}、e: {choice4}
回答: | [abcde] |
| 5-0 | Answer the question

{question}

Answer using only the leading number of one of the following options.
0: {choice0}, 1: {choice1}, 2: {choice2}, 3: {choice3}, 4: {choice4}
Answer: | 質問に回答してください。

{question}

次の選択肢から先頭の数字のみを選択して回答してください。
0: {choice0}、1: {choice1}、2: {choice2}、3: {choice3}、4: {choice4}
回答: | [0-4] |
| 5-1 | Answer the question

{question}

Answer using only the leading letter of one of the following options.
a: {choice0}, b: {choice1}, c: {choice2}, d: {choice3}, e: {choice4}
Answer: | 質問に回答してください。

{question}

次の選択肢から先頭の英字のみを選択して回答してください。
a: {choice0}、b: {choice1}、c: {choice2}、d: {choice3}、e: {choice4}
回答: | [abcde] |

Table 19: The evaluation templates for CommonsenseQA and JCommonsenseQA (Part 2 of 2).