# OpenReview forum: "Toward the Evaluation of Large Language Models Considering Score Variance across Instruction Templates"
_EMNLP/2024/Workshop/BlackBoxNLP — BlackboxNLP 2024_

### Official Review · Reviewer_SASg · 2024-08-25

**Overall Assessment:** 2
**Confidence:** 4

**Best Paper:**

1

**Best Paper Justification:**

N/A

**Comments Questions Suggestions And Typos:**

- From lines 71 to 74, I couldn't follow the statements about examining the effect of knowledge transfer. Why jump to this? I believe the sentences above are mostly about how you develop new metrics based on various evaluation templates.
 - For line 85, I can't open the anonymous link - https://anonymized_for_review/
(minor one)
- I think this question is not only restricted to one language. Why did you choose Japanese as the complement to that? Is it due to the convenience of existing Japanese datasets like JGLUE, which you can easily use for experiments?

**Paper Summary:**

This paper essentially investigates the robustness of evaluation from the perspective of sensitivity to prompt perturbations, i.e., how LLMs' performance scores are sensitive to different prompts. However, the authors spend 6.5 pages discussing the fine-tuning setup, English/Japanese language setup, and even HuggingFace model names (Fig. 2). It's fair to say that putting this information into an appendix would be a more appropriate choice. Overall, I believe this work is currently not suitable for the conference and has much room for improvement.

**Summary Of Strengths:**

- The research question is very important and represents an underexplored research area.
- The number of conducted experiments is sufficient for a main conference.

**Summary Of Weaknesses:**

The primary strength of this paper is that it addresses an interesting and important research question. However, the author fails to design effective experiments to answer this question. The critical RQ for this paper lies with the metrics people commonly used to evaluate IT. The first question to address is: how fragile are these traditional accuracy-based metrics, and what is the scale of variance for them? Based on that, we could identify the potential flaws when we do not consider prompt differences. The research community would likely not be overly concerned with whether the model used to prove the previous assumptions is in English, Japanese, or other languages, as long as a reasonable variety is included. Therefore, I would encourage the authors to reorganize the structure of the project, highlighting the exhaustiveness of the prompt perturbation experiments conducted, and validate the new metrics used (based on $\mu$ and $\sigma^2$ in this paper).

---

### Official Review · Reviewer_A8j8 · 2024-09-06

**Overall Assessment:** 4
**Confidence:** 5

**Best Paper:**

1

**Best Paper Justification:**

-

**Comments Questions Suggestions And Typos:**

Q: in figure 1 i see minor changes in the different evaluation templates (numbers replaced by letters for the option list ofr instance). Could more important changes be considered as well ?
Q: what is Matthews correlation coefficient (MCC) ?
Q: What about generating the multiple tempates automatically (for instance suing GPYT-4) ?

**Paper Summary:**

The paper investigates the LLMs' sensitivity to prompt variation in NLU. While instruction tuning over diverse prompt templates and tasks can reduce this variability, current evaluation frameworks typically assess models using only a single template, ignoring performance variation across different prompts. Authors’ goal is thus to better evaluate models' generalization capabilities by taking into account performance across multiple instruction templates. For this, a new dataset (containing multiple evaluation instructions) and a new metric (Sharpe score, which accounts for the variability across prompts) are proposed and experimented with in zero-shot and full-fine-tuning (NLU) settings.

**Summary Of Strengths:**

-LLMs' sensitivity to prompt variation in NLU is understudied and this paper brings contribution to this topic

-it also provides interesting insights on using plain vs instructed models in zero-shot NLU

-it systematically compares decoding methods (greedy+postproc vs contrained decoding) and advocates for constrained decoding when evaluating the zero-shot setting

-a scalable metric that takes into account and penalize models with high variance of results across templates, is proposed

**Summary Of Weaknesses:**

Fig 1 is not crystal clear: could be improved and more commented, more generally unclear if the generation of mutiple prompt templates is done manually or automatically (for isntance using gpt-4 or anotehr strong model)

Would have been good to see the topline performance of a strong model in zero-shot here (gpt-4o for instance)

---

### Decision · Program_Chairs · 2024-09-20

**Decision:**

Accept

**Comment:**

The reviewers appreciate the topic of the paper, and particularly reviewer A8j8 considers it a strong contribution to BlackboxNLP. We encourage the authors to take this reviewers comments into account for the camera ready version of the paper.